# Evaluating Selective Encryption Against Gradient Inversion Attacks

## Abstract

Gradient inversion attacks pose significant privacy threats to distributed training frameworks such as federated learning, enabling malicious parties to reconstruct sensitive local training data from gradient communications between clients and an aggregation server during the aggregation process. While traditional encryption-based defenses, such as homomorphic encryption, offer strong privacy guarantees without compromising model utility, they often incur prohibitive computational overheads. To mitigate this, selective encryption has emerged as a promising approach, encrypting only a subset of gradient data based on the data's significance under a certain metric. However, there have been few systematic studies on how to specify this metric in practice. This paper systematically evaluates selective encryption methods with different significance metrics against state-of-the-art attacks. Our findings demonstrate the feasibility of selective encryption in reducing computational overhead while maintaining resilience against attacks. We propose a distance-based significance analysis framework that provides theoretical foundations for selecting critical gradient elements for encryption. Through extensive experiments on different model architectures (LeNet, CNN, BERT, GPT-2) and attack types, we identify gradient magnitude as a generally effective metric for protection against optimization-based gradient inversions. However, we also observe that no single selective encryption strategy is universally optimal across all attack scenarios, and we provide guidelines for choosing appropriate strategies for different model architectures and privacy requirements.

## 1 Introduction

As ever-larger machine learning models need to be trained on ever-larger amounts of data, e.g., large language models (LLMs) or computer vision models (Verbraeken et al., 2020), *distributed* training methods have become essential to many model training pipelines. Typically, such training methods allow multiple clients, e.g., separate GPUs, to each hold a subset of the training data; each client then computes updates to a local copy of the model based on its local training data. These model copies are aggregated at a coordinator server to obtain a global model that is then sent back to the clients, and this process repeats in an iterative manner. Popular variants of this distributed training framework include federated learning, e.g., FedSGD and FedAvg (McMahan et al., 2017), in which the frequency of model aggregations is reduced in order to reduce communication overhead or preserve the privacy of client data.

While privacy is often cited as one of the key benefits of a federated learning framework, as raw training data never leaves the clients (Li et al., 2020), federated learning still raises privacy risks. In particular, raw model gradients are communicated between the clients and the coordinator server, which can be used to reconstruct client data via *gradient inversion attacks* (Zhang et al., 2022). Recent studies have demonstrated that these attacks can achieve near-perfect reconstruction of training data in certain scenarios, making gradient privacy a critical concern for real-world federated learning deployments (Geiping et al., 2020; Balunovic et al., 2022; Petrov et al., 2024).

Several works have accordingly proposed modifications to the typical federated learning framework that aim to enhance privacy, e.g., differential privacy (Wei et al., 2020) and secure aggregation (Bonawitz et al., 2016).

Homomorphic encryption, in which clients encrypt their model updates before sending them to the central server, where they are aggregated without being decrypted, is often preferable as it easily accommodates client dropouts, unlike secure aggregation, and it does not slow down convergence with respect to training rounds, unlike differential privacy (Jin et al., 2023). However, homomorphic encryption introduces high computing and communication overheads. For instance, homomorphic encryption can increase communication costs by 10-100x and computational overhead by similar factors, making it impractical for client devices that are resource-constrained, as is typical in federated learning scenarios (Jin et al., 2023). Thus, prior work has suggested selective or partial gradient encryption approaches (Figure 1) to reduce computational overhead while preserving data privacy (Jin et al., 2023).

Selective gradient encryption methods, inspired by the selective encryption methods that have been applied in other domains (Spanos & Maples, 1995), aim to select a subset of model gradients to encrypt in each communication round. By encrypting only the gradients that carry the most information about the training data on which they are computed, we ensure that gradient inversion attacks cannot reconstruct this training data. However, while several prior works have proposed different methods for measuring the significance of different gradient parameters, many are either based on heuristics or only evaluated on smaller models, leaving the open question of *which gradient significance measure best defends against gradient inversion attacks.* Complicating this choice further, some metrics like *sensitivity* may require significant additional computation, making them impractical (McRae et al., 1982). In this paper, we fill this research gap by

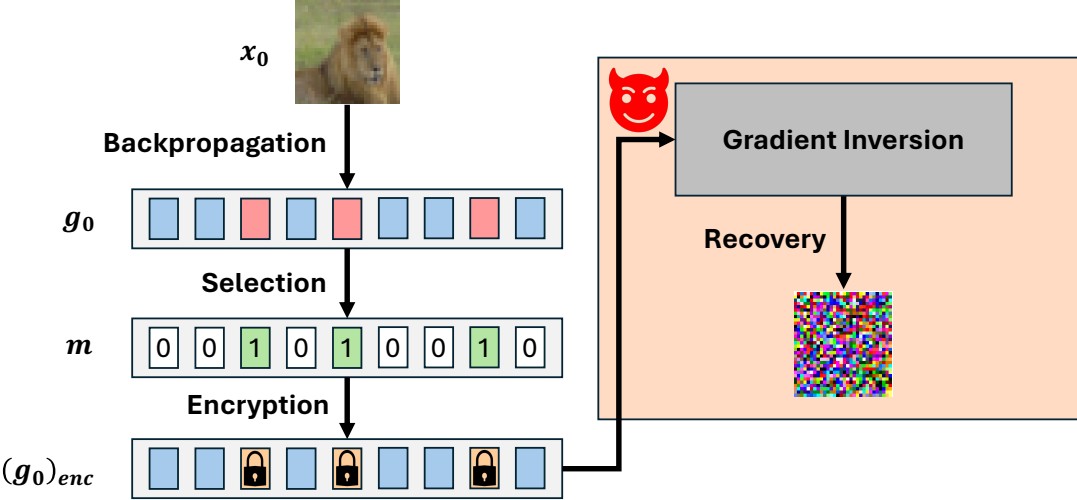

Figure 1: Selective Encryption Against Gradient Inversion Attacks. The gradient $g_0$ calculated by the ground-truth data $x_0$ is selectively encrypted based on the mask $m$ generated according to some significance metric. Red indicates elements with high significance. The adversary can try to recover $x_0$ by matching the generated gradient with the eavesdropped encrypted gradient $(g_0)_{enc} := (1 - m) \odot g_0 + E(m \odot g_0)$.

providing the first comprehensive evaluation framework, both theoretically and empirically, for selective encryption strategies against gradient inversion attacks. Our evaluation spans multiple model architectures (LeNet, CNN, BERT, GPT-2), attack types, and selection measures. Our main contributions are as follows:

- We are the first to **formally describe the selective encryption procedure with different significance metrics** and **systematically evaluate its effectiveness** against gradient inversion attacks.

- We propose a **distance-based significance analysis framework** to analyze selective encryption's ability to defend against gradient inversion attacks, and we use this analysis to propose a new, distance-based significance measure.

- We **evaluate the effectiveness of selective encryption with different significance metrics** in defending gradient inversion attacks for distributed learning.

- Based on our evaluation results, we **provide general guidelines for how to choose defense strategies** for different learning tasks.

We first give an overview of related work in Section 2 before introducing our threat model in Section 3. We present our analysis of selective encryption methods in Section 4 and our experimental evaluation of these methods in Section 5 before discussing future work in Section 6 and concluding in Section 7.

## 2 Background and Related Work

### 2.1 Gradient Inversion Attacks

In the rest of the paper, we denote $\boldsymbol{x_0}, \boldsymbol{x^*} \in \mathbb{R}^n$ as the ground truth and recovered input data respectively, $\boldsymbol{y_0}$, $\boldsymbol{y^*} \in \mathbb{R}^k$ as the corresponding one-hot vector representing the ground truth and recovered label respectively, $\boldsymbol{\theta} \in \mathbb{R}^m$ as the model parameters, $\boldsymbol{f}(\boldsymbol{x}, \boldsymbol{\theta}) : \mathbb{R}^n \times \mathbb{R}^m \to \mathbb{R}^k$ as the machine learning model, $\mathcal{L} : \mathbb{R}^n \times \mathbb{R}^k \times \mathbb{R}^m \to \mathbb{R}$ as the model's loss function, and $\boldsymbol{g} := \nabla_{\boldsymbol{\theta}} \mathcal{L}$ as the gradient of the loss with respect to $\boldsymbol{\theta}$. For simplicity, in the following sections, we may write $\boldsymbol{g_0}(\boldsymbol{\theta}) := \boldsymbol{g}(\boldsymbol{x_0}, \boldsymbol{\theta})$ as the ground truth gradient data evaluated with the loss at $\boldsymbol{x_0}$ and $\boldsymbol{\theta}$, and $\boldsymbol{g^*}(\boldsymbol{\theta}) := \boldsymbol{g}(\boldsymbol{x^*}, \boldsymbol{\theta})$ as the gradient data reproduced by the attacker with reconstructed training data $\boldsymbol{x^*}$ using its knowledge of the model structure and loss function $\mathcal{L}$.

The general procedure for gradient inversion attacks usually includes the formulation of the optimization problem

$$\arg\min_{\boldsymbol{x}, \boldsymbol{y}} \|\boldsymbol{g}(\boldsymbol{x}, \boldsymbol{y}, \boldsymbol{\theta}) - \boldsymbol{g}(\boldsymbol{x_0}, \boldsymbol{y_0}, \boldsymbol{\theta})\| + \alpha \mathcal{R}(\boldsymbol{x}, \boldsymbol{y}), \tag{1}$$

where the first term shows the effort to match the gradient generated by recovered data with the ground truth, and $\alpha \mathcal{R}(\boldsymbol{x}, \boldsymbol{y})$ is a regularization term varying for different methods (Huang et al., 2021). Deep Leakage (Zhu et al., 2019), Inverting Gradients (Geiping et al., 2020), and TAG (Deng et al., 2021) follow this framework with variations in the regularization term $\mathcal{R}(\boldsymbol{x}, \boldsymbol{y})$ or the norm. Other approaches, such as Decepticons (Fowl et al., 2022), DAGER (Petrov et al., 2024), and APRIL (Lu et al., 2022), incorporate analytical methods tailored to specific model types, achieving remarkable recovery performance with high efficiency. Recent studies further enhance recovery capabilities by integrating auxiliary models (Balunovic et al., 2022; Yue et al., 2023). Despite the development of various sophisticated techniques, the general optimization-based framework remains the foundation and starting point of these methods.

In response to the gradient inversion attacks, researchers have also proposed various defense methods. Gradient pruning (Zhang et al., 2023) demonstrates the defense effectiveness of pruning the large gradient elements but does not provide any theoretical support. The use of differential privacy is lightweight but leads to inevitable downgrading in the model utility (Wei et al., 2020), and works such as Yue et al. (2023) also show the vulnerability of naive differential privacy mechanisms. Other defense frameworks include Soteria (Sun et al., 2021), ATS (Gao et al., 2021), and PRECODE (Scheliga et al., 2022), though these are shown to be sometimes unreliable by works like Balunović et al. (2021). Encryption-based defenses have high reliability and thus are still of interest, but have high computational overhead.

### 2.2 Sensitivity Analysis

Existing works on sensitivity analysis of model gradients provide the foundation for the selection of encrypted parameters in the efficient federated learning system proposed by Jin et al. (2023). Novak et al. (2018) introduces two sensitivity metrics, the Frobenius norm of the input-output Jacobian and the number of linear region transitions, for fully connected neural networks. Yeh et al. (2019) developed objective evaluation measures for machine learning explanations, including infidelity (how well explanations capture model behavior under perturbations) and sensitivity (explanation stability under small input changes). Despite these advances in sensitivity analysis frameworks, there remains a gap in the literature regarding the use of these frameworks to defend against gradient inversion attacks specifically. Though Mo et al. (2021) propose an empirical sensitivity analysis framework, their layer-wise analysis is coarse and is not evaluated against gradient inversions. To our knowledge, existing approaches do not adequately explain how different parameters affect gradient-based data recovery results.

### 2.3 Selective Encryption

Existing selective encryption methods are mainly proposed to reduce the delay and encrypted data size during real-time video transmission over potentially insecure networks (Spanos & Maples, 1995; Kunkelmann & Reinema, 1997). Their selection strategies rely on techniques like data compression and Discrete Cosine Transform (DCT) coefficients, with choices primarily guided by ad hoc evaluations. These are difficult to extend to our scenario of using selective encryption to defend models against gradient inversion attacks.

In recent years, selective encryption has been applied to the regime of machine learning to enhance security while efficiently handling the growing size of models. Tian et al. (2021) proposes a probabilistic selection strategy to encrypt the parameters of Convolutional Neural Networks (CNNs). Sparse Fast Gradient Encryption (SFGE) identifies the most critical NN weights to encrypt based on their sensitivity to adversarial perturbations (Cai et al., 2019). However, it does not address the threat of gradient inversion attacks, focusing solely on protecting the model parameters from being extracted. Zuo et al. (2021) focuses on the convolution layers in CNNs and adopts the $\ell_1$-norm of each kernel row as the significance metric. However, these efforts are limited by their reliance on specific model structures, and to the best of our knowledge, no study has systematically compared the effectiveness of selective encryption methods against gradient inversion attacks for more general types of neural network models, as we do in this work.

## 3 Threat Model

We restrict the attacking scenario to be *honest-but-curious*, where the adversary $\mathcal{A}$ can obtain clients' model gradients through eavesdropping when they are communicated to the server or corrupting the aggregation server during distributed training. The attack setting is assumed to be white-box, that is, $\mathcal{A}$ always knows the model structure, the current global model parameter values, as well as the gradient used to update them at an arbitrary training iteration. The training setting assumes stochastic sampling of the clients and their data points across iterations.

We also assume that $\mathcal{A}$ is strong enough so that all gradient elements except those encrypted can be perfectly reproduced. As encryption methods typically transform data into seemingly random or nonsensical patterns, we assume that an attacker can easily identify the encrypted portion of the gradient elements and exclude them during the reconstruction process. This filtering prevents the attacker from being misled by the encrypted content, allowing them to perform the reconstruction relying solely on the meaningful, unencrypted gradient elements. This also means that the elements of generated gradient $\boldsymbol{g^*}(\boldsymbol{\theta_0})$ coincide with those of the ground truth gradient $\boldsymbol{g_0}(\boldsymbol{\theta_0})$ at every position except for those in the set $M$ of indices of masked gradient elements, while we claim that the other elements of $\boldsymbol{g^*}(\boldsymbol{\theta_0})$ are random and bounded in a certain range. More formally, we assume that: $\boldsymbol{g^*}(\boldsymbol{\theta_0})^{(i)} = \boldsymbol{g_0}(\boldsymbol{\theta_0})^{(i)}$ for any $i \notin M$ and otherwise, $\boldsymbol{g^*}(\boldsymbol{\theta_0})^{(i)} = \delta_i$, where $\|\delta_i\| \leq \xi$ for some $\xi > 0$.

Additionally, we assume that $\mathcal{A}$ has access to the ground truth label $\boldsymbol{y_0}$ throughout this study. This assumption is based not only on prior works (Geiping et al., 2020; Geng et al., 2021; Yin et al., 2021), which demonstrate the feasibility of label recovery, but also on our intent to evaluate the attack in a worst-case scenario where the attacker has as much information as possible.

## 4 Selective Encryption Based on Significance Metrics

In this section, we formalize the selective encryption process and give an overview of the metrics that we later evaluate. We then present our distance-based analysis, which provides theoretical foundations for selecting gradient elements most critical to data privacy.

## 4.1 Selective Encryption

Let $\boldsymbol{g}$ represent the gradient of a model's parameters at a given training step. We express $\boldsymbol{g}$ as a vector of $\mathbb{R}^m$ where $m$ is the number of model parameters:

$$\boldsymbol{g} = \left(\boldsymbol{g}^{(1)}, \boldsymbol{g}^{(2)}, \ldots, \boldsymbol{g}^{(m)}\right)^T. \tag{2}$$

To protect against gradient inversion attacks, we use a given significance metric to attach a significance value to each gradient element $\boldsymbol{g}^{(i)}$, $i = 1, 2, \ldots, m$. We then encrypt the gradient elements with the top $s$ significance values. We define the encryption function $\boldsymbol{E}$ applied to a subset of gradients, parameterized by a binary mask $\boldsymbol{m} \in \{0, 1\}^m$:

$$\boldsymbol{E}(\boldsymbol{g}, \boldsymbol{m}) = \boldsymbol{E}(\boldsymbol{m} \odot \boldsymbol{g}), \tag{3}$$

where $\boldsymbol{m}_i = 1$ indicates the selection of the $i$-th element to encrypt. We also denote $M := \{i | \boldsymbol{m}_i = 1\}$ as the set of selected indices.

## 4.2 Overview of Significance Metrics

We summarize the significance metrics considered in Table 1. We consider sensitivity-based and distance-based metrics as two fundamentally different approaches to identifying critical gradient components for selective encryption. The sensitivity (**Sens**) metric is adopted from previous studies of selective encryption against gradient inversion, where the second-order derivative of the gradient with respect to the model inputs, $\nabla_{\boldsymbol{x_0}} \nabla_{\boldsymbol{\theta}} \mathcal{L}$, serves as the sensitivity measure (Jin et al., 2023). This is a natural choice given the iterative nature of data reconstruction in gradient inversions, assuming that gradient components most susceptible to input perturbations are the most informative for attackers and thus should be prioritized for protection. The product significance (**ProdSig**) and gradient magnitude (**Grad**) metrics are derived from our distance-based analysis presented in the following section, which assigns more significance to the geometric contribution of gradient elements to the reconstruction process itself. We also examine model parameter magnitude (**Param**) primarily due to its ease of access and implementation.

Table 1: Significance Metrics to Evaluate. Note that **Grad** and **Param** Require No Additional Computation ("Get for Free").

| Metric | Sensitivity-Based | Distance-Based | Get for Free |
|---|---|---|---|
| Sensitivity (**Sens**) | ✓ | ✗ | ✗ |
| Product Significance (**ProdSig**) | ✗ | ✓ | ✗ |
| Gradient (**Grad**) | ✗ | ✓ | ✓ |
| Model Parameter (**Param**) | ✗ | ✗ | ✓ |

## 4.3 Distance-Based Significance Analysis

We next aim to maximize the distance between recovered and ground-truth data through selective encryption, focusing primarily on tasks employing the widely-used cross-entropy loss. Our theoretical framework builds on the observation that gradient inversion attacks succeed by minimizing the distance between reconstructed and original gradients. By selectively encrypting the gradient elements that most strongly influence this distance, we can effectively disrupt the attack while maintaining computational efficiency.

Leveraging Lipschitz continuity assumptions, we derive two effective metrics: Product Significance (ProdSig), which utilizes the product of gradient elements and model parameters, and Gradient Magnitude (Grad), which employs the absolute values of gradient elements. These metrics establish theoretical lower bounds on the distance between recovered and ground truth data, providing principled approaches to selective encryption that advance beyond existing sensitivity-based methods.

The theoretical foundation rests on the following key insight: if we can ensure that the most informative gradient elements (those that contribute most to reducing reconstruction distance) are encrypted, we can establish a lower bound on the achievable reconstruction quality by any gradient inversion attack.

Beginning with several Lipschitz continuity-based assumptions, we derive Lemma 4.3 (proof in Appendix A.2), which serves as the foundation for our comprehensive distance-based analysis.

**Assumption 4.1.** $L(\boldsymbol{x}, \boldsymbol{\theta})$ is Lipschitz continuous with a coefficient $C_L$ with respect to the input $\boldsymbol{x}$.

*Justification*: This assumption holds for most practical loss functions used in machine learning, including cross-entropy loss with bounded inputs.

**Assumption 4.2.** $\boldsymbol{g}(\boldsymbol{x}, \boldsymbol{\theta})$ is smooth with respect to the model parameter $\boldsymbol{\theta}$ and vary slowly near $\boldsymbol{\theta}$. More formally, given any $t \in [0, 1]$, $\|\boldsymbol{g}(\boldsymbol{x}, t\boldsymbol{\theta}) - \boldsymbol{g}(\boldsymbol{x}, \boldsymbol{\theta})\| \leq \xi$ for some $\xi > 0$.

*Justification*: This assumption is reasonable for neural networks with continuous activation functions, where small changes in parameters lead to small changes in gradients in local neighborhoods.

**Lemma 4.3.** *Under Assumptions 4.1 and 4.2,*

$$\log\left(\boldsymbol{f}(\boldsymbol{x}, \boldsymbol{\theta})^{(k_0)}\right) \approx -\sum_{i=1}^{m} \boldsymbol{g}(\boldsymbol{x}, \boldsymbol{\theta})^{(i)} \boldsymbol{\theta}^{(i)} + C, \tag{4}$$

*where* $C := \log\left(\boldsymbol{f}(\boldsymbol{x_0}, \boldsymbol{0})\right) = \log\left(\boldsymbol{f}(\boldsymbol{x^*}, \boldsymbol{0})\right)$ *indicates the output of an all-zero model.*

Under Assumption 4.1, the distance-based significance analysis is formulated as

$$\|\boldsymbol{x^*} - \boldsymbol{x_0}\| \geq \frac{1}{C_L} \left| \log\left(\boldsymbol{f}(\boldsymbol{x^*}, \boldsymbol{\theta})^{(k_0)}\right) - \log\left(\boldsymbol{f}(\boldsymbol{x_0}, \boldsymbol{\theta})^{(k_0)}\right) \right| \tag{5}$$

$$\approx \frac{1}{C_L} \left| \sum_{i=1}^{m} \boldsymbol{g^*}(\boldsymbol{\theta})^{(i)} \boldsymbol{\theta}^{(i)} - \sum_{i=1}^{m} \boldsymbol{g_0}(\boldsymbol{\theta})^{(i)} \boldsymbol{\theta}^{(i)} \right| \tag{6}$$

$$= \frac{1}{C_L} \left| \sum_{i \in M} \left(\boldsymbol{g^*}(\boldsymbol{\theta})^{(i)} - \boldsymbol{g_0}(\boldsymbol{\theta})^{(i)}\right) \boldsymbol{\theta}^{(i)} \right|, \tag{7}$$

where the Approximation 6 holds under Lemma 4.3. Equation 7 is then dominated by $\left| \sum_{i \in M} \boldsymbol{g_0}(\boldsymbol{\theta})^{(i)} \boldsymbol{\theta}^{(i)} \right| \leq \sum_{i \in M} \left| \boldsymbol{g_0}(\boldsymbol{\theta})^{(i)} \boldsymbol{\theta}^{(i)} \right|$ due to the claim that $\boldsymbol{g^*}(\boldsymbol{\theta})^{(i)}$ is bounded. This shows that the product of masked $\boldsymbol{g_0}(\boldsymbol{\theta})^{(i)}$ and $\boldsymbol{\theta}^{(i)}$ can serve as an indicator of the distance between the recovered and ground truth data. We name it as **Product Significance (ProdSig)**.

**Assumption 4.4.** $\boldsymbol{g}(\boldsymbol{x}, \boldsymbol{\theta})$ is Lipschitz continuous with a coefficient $C_g$ with respect to the input $\boldsymbol{x}$.

Similarly, under Assumption 4.4, we use Equation (7) to obtain

$$\|\boldsymbol{x^*} - \boldsymbol{x_0}\| \geq \frac{1}{C_g} \|\boldsymbol{g^*}(\boldsymbol{\theta}) - \boldsymbol{g_0}(\boldsymbol{\theta})\|, \tag{8}$$

and the bound is dominated by $|\boldsymbol{g_0}(\boldsymbol{\theta})^{(i)}|$ for $i \in M$. Therefore, we claim the magnitude of gradient elements can also be seen as a significance indicator, which we call **Grad** in the following sections.

**Key Theoretical Findings.**

- Distance-based analysis establishes rigorous theoretical lower bounds on reconstruction error, providing formal guarantees on defense effectiveness.

- Product Significance (ProdSig) maximizes the theoretical bound on the data reconstruction error through products of gradient and parameter values, offering optimal protection under our assumptions.

- Gradient magnitude (Grad) offers a computationally efficient yet theoretically sound alternative to ProdSig that requires no additional computation.

**Practical Implications.** Our theoretical analysis provides clear guidance for practitioners: When computational resources are abundant, ProdSig offers theoretically optimal protection, while Grad provides near-optimal protection with zero computational overhead, making it ideal for resource-constrained federated learning scenarios.

# 5 Experiments

In this section, we aim to illustrate the defense effectiveness and efficiency of the significance metrics we consider. In addition to those presented in Table 1, we also take into account the magnitude of model parameters (**Param**) as an intuitive way to indicate the significance. For large attention-based models, we also evaluate encryption of the attention layers (**Attn**) and one-fifth of the vulnerable layers (**OneFifth-i**), which we would intuitively expect to carry significant information.

## 5.1 Experimental Setup

**Tasks.** We evaluate our defense methods against attacks on fundamental machine learning tasks, including image and sequence classifications, demonstrating their effectiveness against strong adversaries. To assess performance in the worst-case scenario, we set the batch size to 1, i.e., each gradient is computed on a single data point.

**Models.** We test our framework on vision models including LeNet (Lecun et al., 1998) (88,648 parameters) and a CNN with two $5 \times 5$ convolution layers (McMahan et al., 2017) (2,202,660 parameters) and language models including BERT (Devlin et al., 2019) (109,483,778) and GPT-2 (Radford et al., 2019) (124,441,344 parameters).

**Attacks.** For image models, we use the Inverting Gradients (IG) (Geiping et al., 2020), as it shows stable attacking outcomes. For language models, we use the recently proposed optimization-based attack LAMP (Balunovic et al., 2022) and the analytical attack DAGER (Petrov et al., 2024), which is claimed to attack large language models successfully.

**Metrics.** Building on previous studies evaluating gradient inversions on images (Huang et al., 2021; Yin et al., 2021), we use the Learned Perceptual Image Patch Similarity (LPIPS) introduced by Zhang et al. (2018) to quantify image dissimilarity, where higher values indicate greater differences (i.e., reconstructed images that are less similar to the ground truth images). In addition, we report the Mean Squared Error (MSE) between the reconstructed and ground truth data to highlight the impact of our distance-based analysis. For language models, we evaluate recovery performance using the ROUGE-1 metric proposed by Lin (2004), while the Wasserstein distance is used to measure differences between embeddings. We further evaluate the additional time needed to calculate each selective encryption method, which quantifies its computational overhead.

**Datasets.** We use the image classification dataset CIFAR-100 (Krizhevsky, 2012) for vision tasks, and sentiment analysis datasets CoLA (Warstadt et al., 2019), SST-2 (Socher et al., 2013), and Rotten Tomatoes (Pang & Lee, 2005) for language tasks.

## 5.2 Defense Effectiveness

We use 10 random samples (see Figure 2) from the CIFAR-100 dataset for the evaluation of Inverting Gradients, 5 sequences from each of the sequence classification datasets for LAMP, and 10 sequences for DAGER. We perform at least 5 repetitions on each sample and take the best recovery results. The averaged best recovery results among the samples are then presented in the figures in Appendix A.4.

We summarize the minimal encryption ratio needed for each method to reach a desired protection level in Table 4, where NA means this method fails to reach the level within the range we choose. We empirically set the desired protection level to LPIPS $\geq 0.3$ for image tasks and ROUGE-1 $\leq 0.2$ for language tasks. We also summarize the distance between the recovered and ground-truth data for each case in Table 5, where we fix the encryption ratio at 30% to make comparisons before the convergence of recovery and avoid the unexpected distance drop due to the appearance of nearly blank recovered images.

**Summary of Findings.** Our comprehensive evaluation across multiple models and attack types reveals distinct performance patterns for each significance metric. **Grad** emerges as the most consistent performer, while other metrics show scenario-specific strengths. **Sens** excels against analytical attacks but struggles with LAMP, and **Param** shows surprising effectiveness against DAGER despite failing against optimization-

based attacks. Table 2 provides a qualitative overview of each metric's performance to guide selection based on specific requirements and attack scenarios. Our criteria are:

- **Poor**: Fails to reach desired protection with average $< 60\%$ encryption

- **Moderate**: Reaches desired protection with average 50-60% encryption

- **Good**: Reaches desired protection with average 10-50% encryption

- **Excellent**: Reaches desired protection with average $\leq 10\%$ encryption

Table 2: Qualitative Summary of Significance Metrics Performance Against Different Attacks

| Attack | Model | Sens | ProdSig | Grad | Param |
|--------|-------|------|---------|------|-------|
| IG | LeNet | ★★ | ★★ | ★★ | × |
| | CNN | ★★ | ★★ | ★★ | × |
| LAMP | BERT | × | ★ | ★★ | × |
| | GPT-2 | × | ★★ | ★★★ | × |
| DAGER | GPT-2 | ★★★ | ★★★ | ★★★ | ★★★ |

$^\times$ Poor
$^\star$ Moderate
$^{\star\star}$ Good
$^{\star\star\star}$ Excellent

**Attn and OneFifth.** We summarize the performance of Attn and OneFifth-i against LAMP in Table 3. The performance against DAGER is not reported here as it cannot recover anything if its target attention layers are encrypted. Note that the encryption ratio of OneFifth-i is not exactly 0.2 because the embedding layer is not used and thus not considered for encryption during the iterative reconstruction of LAMP. The results show that encrypting only the attention layers is insufficient, even though they contain much important information, and that the second one-fifth portion of the GPT-2 (ignoring the embedding layer) seems to carry more significance to gradient inversion adversaries.

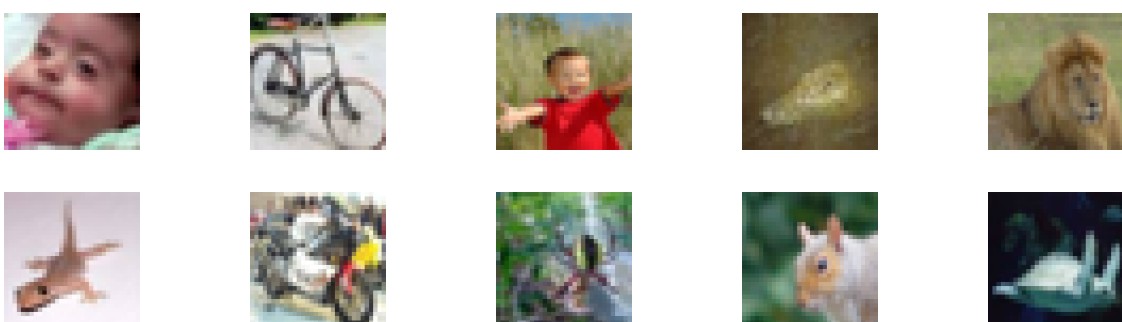

Figure 2: Target Images

**Sens.** The selection by sensitivity shows a stable and (nearly) optimal defense performance against Inverting Gradients and DAGER. However, in some cases against LAMP, encrypting the gradient elements with the highest sensitivity fails to provide sufficient protection with less than 60% of elements encrypted (see Figure 7 in Appendix A.4).

**ProdSig.** Our results show that the product significance acts as a stable significance metric in the defenses against the attacks we consider. As shown in Tabel 4, it provides the second-best protection under Inverting

Table 3: LAMP Recovers the Training Data on GPT-2 Under Attn and OneFifth (CoLA) Selective Encryption.

| Defense | Enc.Ratio | EmbedDist | ROUGE-1 | ROUGE-2 |
|---------|-----------|-----------|---------|---------|
| Attn | 0.2278 | 42.7297 ± 20.4964 | 0.7090 ± 0.1860 | 0.4111 ± 0.2960 |
| OneFifth-1 | 0.1392 | 45.8534 ± 20.6135 | 0.7732 ± 0.1536 | 0.5543 ± 0.2474 |
| OneFifth-2 | 0.1519 | 40.7201 ± 20.3857 | 0.7389 ± 0.1625 | 0.4409 ± 0.2982 |
| OneFifth-3 | 0.1519 | 39.2692 ± 20.0363 | 0.7810 ± 0.1623 | 0.4690 ± 0.2706 |
| OneFifth-4 | 0.1519 | 30.9490 ± 13.5714 | 0.8578 ± 0.1049 | 0.6538 ± 0.2453 |
| OneFifth-5 | 0.0949 | 31.2776 ± 14.6672 | 0.8640 ± 0.1032 | 0.7591 ± 0.1790 |

Table 4: **Grad** Requires a Lower Minimum Encryption Ratio Needed to Reach a Desired Protection Level of $\geq 0.3$ for LPIPS on Image Tasks or $\leq 0.2$ for ROUGE-1 for Language Tasks.

| | | | Sens | ProdSig | Grad | Param |
|---|---|---|------|---------|------|-------|
| IG | LeNet | CIFAR100 | 0.3 | **0.2** | **0.2** | NA |
| | CNN | CIFAR100 | 0.3 | 0.3 | **0.2** | NA |
| LAMP | BERT | CoLA | NA | 0.6 | **0.4** | NA |
| | GPT-2 | CoLA | NA | 0.5 | **0.1** | NA |
| | | SST-2 | 0.6 | 0.3 | **0.1** | NA |
| DAGER | GPT-2 | CoLA | 0.005 | 0.04 | 0.01 | **0.002** |
| | | SST-2 | **0.002** | 0.02 | 0.01 | **0.002** |

Gradients and LAMP. In some cases such as Figure 4b in Appendix A.4, it also leads to the largest difference between the recovered and ground truth image.

**Grad.** The encryption by gradient magnitude serves as the best defense strategy against Inverting Gradients and LAMP, and its performance against DAGER is also good. Furthermore, it leads to the largest distance between the recovered and ground truth sequence against LAMP (see Figures 7b, 10b, and 11 in Appendix A.4.

**Param.** According to our results, the naive method of using the magnitude of model parameters fails to protect against optimization-based gradient inversions. This suggests that it may not be a wise choice in most cases. However, it does show excellent protection against DAGER. This could be because DAGER relies on a small subset of gradient in the attention layers where the model parameters with large values tend to concentrate.

Table 5: Grad Yields the Largest Distance Between the Recovered and Ground Truth Data at Encryption Ratio of 30%.

| | | | Sens | ProdSig | Grad | Param |
|---|---|---|------|---------|------|-------|
| IG | LeNet | CIFAR | 0.063 ± 0.049 | **0.14 ± 0.07** | 0.11 ± 0.04 | 0.01 ± 0.01 |
| | CNN | CIFAR | 0.084 ± 0.041 | 0.077 ± 0.032 | **0.087 ± 0.032** | 0.01 ± 0.01 |
| LAMP | BERT | CoLA | 10.06 ± 6.20 | 12.30 ± 8.37 | **16.34 ± 7.29** | 10.64 ± 6.99 |
| | GPT-2 | CoLA | 47.17 ± 21.46 | 49.58 ± 21.15 | **52.30 ± 21.14** | 42.76 ± 20.10 |
| | | SST-2 | 16.22 ± 2.97 | 22.37 ± 1.60 | **25.11 ± 2.34** | 17.09 ± 3.22 |

**Effects During Reconstruction.** We also study the effects of our five selective encryption methods during the iterative reconstruction steps of gradient inversions. Figure 3 shows the effort of each methods to hold

back the convergence of the reconstruction loss on a LeNet image example, under different encryption levels. ProdSig and Grad show the best results in this case. However, this is not the case for all the data points (see more results including those on CNN in Appendix A.5). This suggests a potential future direction for evaluating defenses against gradient inversions in terms of reconstruction loss.

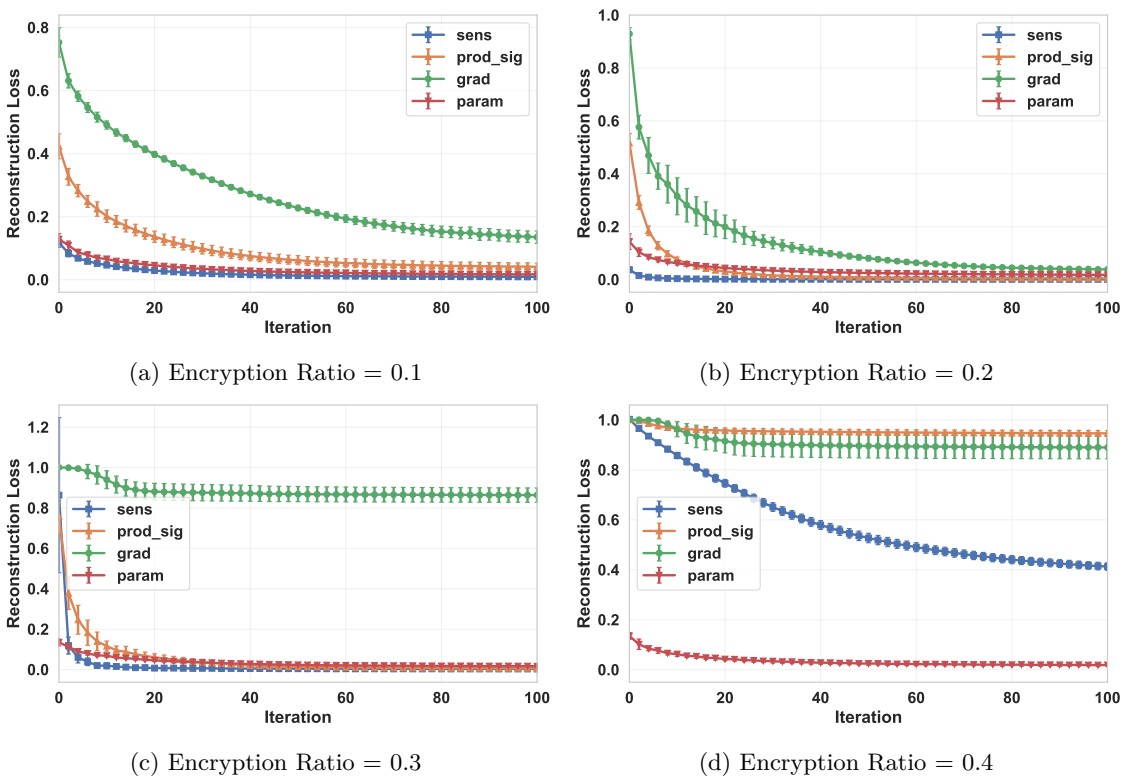

(a) Encryption Ratio = 0.1

(b) Encryption Ratio = 0.2

(c) Encryption Ratio = 0.3

(d) Encryption Ratio = 0.4

Figure 3: Reconstruction Loss During Inverting Gradients on LeNet (Image 1)

Furthermore, increasing the encryption ratio generally improves protection, but at the cost of training efficiency. This trade-off suggests that practical deployments should aim for a balance between encryption strength and model usability. Additionally, model-specific characteristics influence the effectiveness of different encryption strategies, highlighting the need for adaptive approaches tailored to individual architectures.

### 5.3 Defense Efficiency

Finally, we consider calculation time to obtain the significance metrics (Table 6). The magnitude of gradient elements and model parameters can be obtained for free, so they are omitted here. For larger models like BERT and GPT-2, the calculation time of sensitivity can be unacceptably long in our setting, so here we also include the use of a discrete approximation of the real sensitivity, which instead calculates $(g(x + e_i, \theta) - g(x - e_i, \theta))/2$ for each coordinate direction $i$, and takes the mean over all dimensions. It is clear that the calculation of sensitivity takes much longer than the other metrics. Even though discrete approximation greatly reduces the computational overhead of sensitivity, it is still costly for large models like BERT and GPT-2, as shown in Table 6. Without the discrete approximation, sensitivity calculation is not even feasible for larger models. We do not include the gradient method as clients must already calculate gradients, so it does not introduce any additional computational overhead. Since the gradient method is also the best defense method in general (see Section 5.2), we conclude that it is the best choice in most cases.

Table 6: Calculation Time for Significance Metrics (CoLA)

| Model | Layer Num. | Calculation Time (s) | | |
|---|---|---|---|---|
| | | Sens | Sens (Discrete) | ProdSig |
| LeNet | 10 | $80.70 \pm 1.54$ | $11.50 \pm 0.06$ | $0.0009 \pm 0.00006$ |
| CNN | 8 | $1832.13 \pm 20.00$ | $10.01 \pm 0.15$ | $0.0030 \pm 0.00019$ |
| BERT | 201 | - | $3425.18 \pm 1777.28$ | $0.10 \pm 0.27$ |
| GPT-2 | 149 | - | $2669.81 \pm 1610.84$ | $0.07 \pm 0.18$ |

## 6 Discussion

Our comprehensive evaluation across multiple model architectures and attack types provides strong evidence for the practical effectiveness of selective encryption strategies in federated learning scenarios. However, several important considerations, including limitations, emerge from our analysis that warrant discussion for practical deployment.

### 6.1 Limitations

**Still Limited Attack Scenarios.** Our evaluation focused on specific attack methods (Inverting Gradients, LAMP, and DAGER) and model architectures (LeNet, CNN, BERT, and GPT-2). While these represent a range of important benchmarks, they may not capture the full spectrum of possible attack vectors. Sensitivity, for instance, may still offer defensibility through explainability against other gradient-based attacks and their combinations. Future evaluations should consider a broader range of attack methods, particularly as new gradient inversion techniques emerge.

**Protection Level Variability.** The protection levels achieved by our methods vary significantly across different samples, as indicated by the high standard deviations in our results (Table 5). This variability suggests that the effectiveness of our defenses may depend on specific characteristics of the input data or model. Quantitatively characterizing these dependencies may lead to more effective selection encryption defenses tailored to specific input data samples.

### 6.2 Future Work

**Adaptive Defense Strategies.** Future research could explore adaptive defense strategies that dynamically adjust the encryption method and ratio of encrypted gradient elements based on the specific model architecture, data characteristics, and perceived threat level. Such adaptive approaches could optimize the security-efficiency trade-off in different contexts.

**Hybrid Significance Metrics.** Given that different significance metrics show varying effectiveness against different attacks, developing hybrid metrics that combine the strengths of multiple approaches could improve overall defense performance. For instance, a weighted combination of gradient magnitude and product significance might provide more robust protection across a larger range of model architectures and data modalities.

**Scalability Considerations.** Our analysis focuses on single-sample scenarios representing worst-case privacy threats. In practical federated learning with larger batch sizes, selective encryption effectiveness may vary. Future work should investigate how batch size affects the required encryption ratios and whether adaptive strategies can optimize the privacy-utility trade-off in different batch size scenarios.

**Advanced Model Architectures.** Extending our evaluation to more advanced model architectures, particularly larger models and multi-modal architectures, would enhance the practical relevance of our findings in the context of current AI developments.

## 7 Conclusion

This work systematically evaluates different encryption strategies for protecting model gradients from inversion attacks. We find that selecting gradient elements based on their magnitude is the most robust defense against Inverting Gradients and LAMP attacks, while parameter-magnitude-based encryption is highly effective against DAGER. The results also suggest that encrypting attention layers alone is inadequate and that strategic selection of vulnerable model components can yield better protection.

These findings emphasize the importance of adaptive encryption strategies that consider attack type and model architecture. Future work could explore fine-grained layer-wise encryption approaches and assess trade-offs between encryption efficiency and model utility in real-world federated learning scenarios. Additionally, integrating encryption with other privacy-preserving techniques, such as differential privacy or secure multi-party computation, may further enhance defenses against gradient inversion attacks.

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

## A  Appendix

### A.1  Federated Learning

Federated learning was first proposed in McMahan et al. (2017), which builds distributed machine learning models while keeping personal data on clients. Instead of uploading data to the server for centralized training, clients process their local data and share updated local models with the server. Model parameters from a large population of clients are aggregated by the server and combined to create an improved global model.

The FedAvg is commonly used on the server to combine client updates and produce a new global model (McMahan et al., 2017). At each round, a global model $\mathbf{W}_{\text{glob}}$ is sent to $N$ client devices. Each client $i$ performs gradient descent on its local data with $E$ local iterations to update the model $\mathbf{W}_i$. The server then does a weighted aggregation of the local models to obtain a new global model, $\mathbf{W}_{\text{glob}} = \sum_{i=1}^{N} \alpha_i \mathbf{W}_i$, where $\alpha_i$ is the weighting factor for client $i$.

Typically, the aggregation runs using plaintext model parameters through a central server (in some cases, via a decentralized protocol), giving the server visibility of each local client's model in plaintext.

### A.2  Proof of Lemma 4.3

*Proof.* From the definition of the cross entropy, and supposing $k_0$ is the index of the true label, we can write

$$g(x, \theta) = \nabla_{\theta} \mathcal{L}(x, \theta) \tag{9}$$

$$= -y_0^{(k_0)} \nabla_{\theta} \log \left( f(x, \theta)^{(k_0)} \right). \tag{10}$$

Applying the fundamental theorem of calculus then gives

$$\log \left( f(x, \theta)^{(k_0)} \right) - \log \left( f(x, \theta')^{(k_0)} \right) = -\int_{\theta'}^{\theta} g_0(\theta) \cdot \mathrm{d}\theta \tag{11}$$

$$= -\int_{0}^{1} g_0(r(t)) \cdot r'(t) \mathrm{d}t, \tag{12}$$

where $r : \mathbb{R} \to \mathbb{R}^m$ is a curve with $\theta$ and $\theta'$ as its endpoints. Without loss of generality, we take $\theta' = \mathbf{0}$ and $r(\theta) = t\theta$ for $t \in [0, 1]$ and

$$\log \left( f(x, \theta)^{(k_0)} \right) = -\int_{0}^{1} g(x, t\theta) \cdot \theta \mathrm{d}t + C \tag{13}$$

$$= -\sum_{i=1}^{m} \int_{0}^{1} g(x, t\theta)^{(i)} \theta^{(i)} \mathrm{d}t + C \tag{14}$$

$$= -\sum_{i=1}^{m} g(x, t_{\mu}\theta)^{(i)} \theta^{(i)} + C, \tag{15}$$

for some $t_{\mu} \in [0, 1]$. Under Assumption 4.2 that $g(x, \theta)$ varies slowly near $\theta$, we can do the approximation and yield

$$\log \left( f(x, \theta)^{(k_0)} \right) \approx -\sum_{i=1}^{m} g(x, \theta)^{(i)} \theta^{(i)} + C. \tag{16}$$

$\square$

### A.3  Devices Used for Experiments

**Devices.** For the attack experiments, we use (1) Intel 4-core 2.50GHz Xeon Platinum 8259CL CPU with 16GB memory and NVIDIA Tesla T4, (2) Intel 10-core 2.80GHz i9-10900 CPU with 32GB memory and NVIDIA GeForce RTX 3090, and (3) AMD EPYC 7R32 8-core 3.30GHz CPU with 32GB memory and NVIDIA A10G according to the needs. The calculation time measures are all on Intel 2.50GHz Xeon Platinum 8259CL CPU with 16GB memory and NVIDIA Tesla T4.

## A.4 Defense Effectiveness

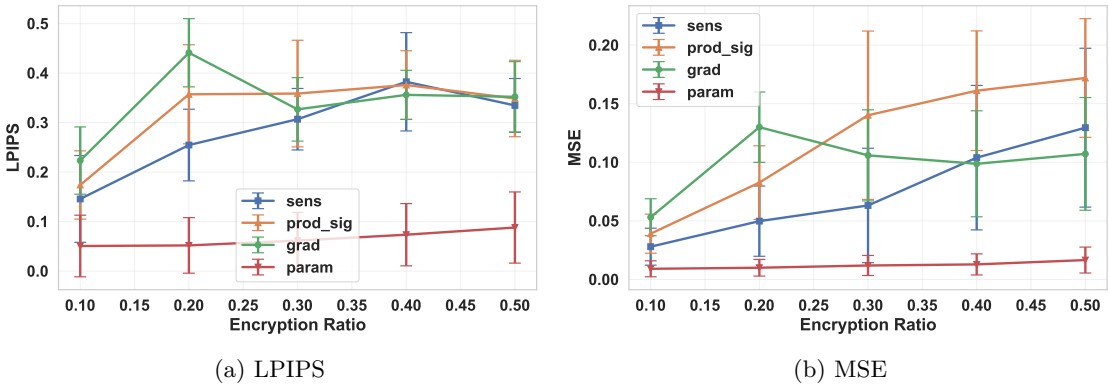

(a) LPIPS

(b) MSE

Figure 4: Inverting Gradients on LeNet Under Defenses

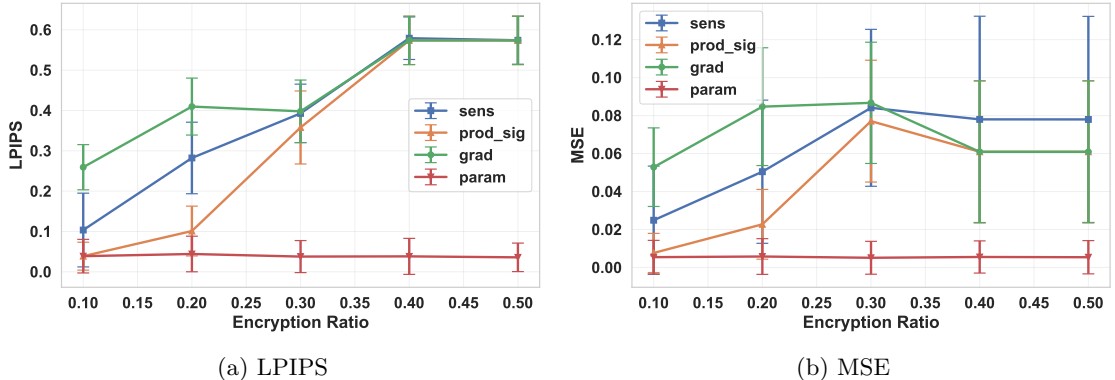

(a) LPIPS

(b) MSE

Figure 5: Inverting Gradients on CNN Under Defenses

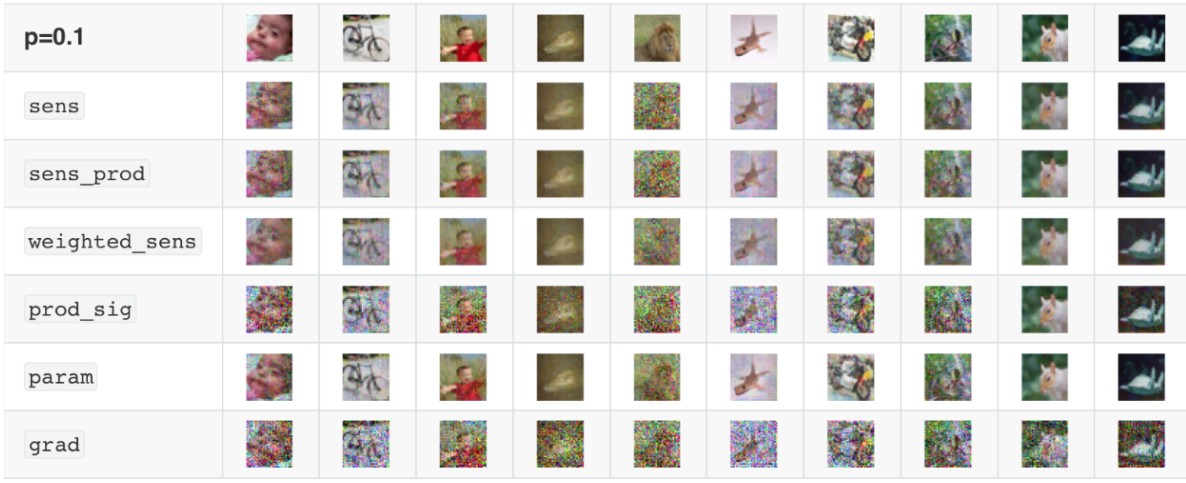

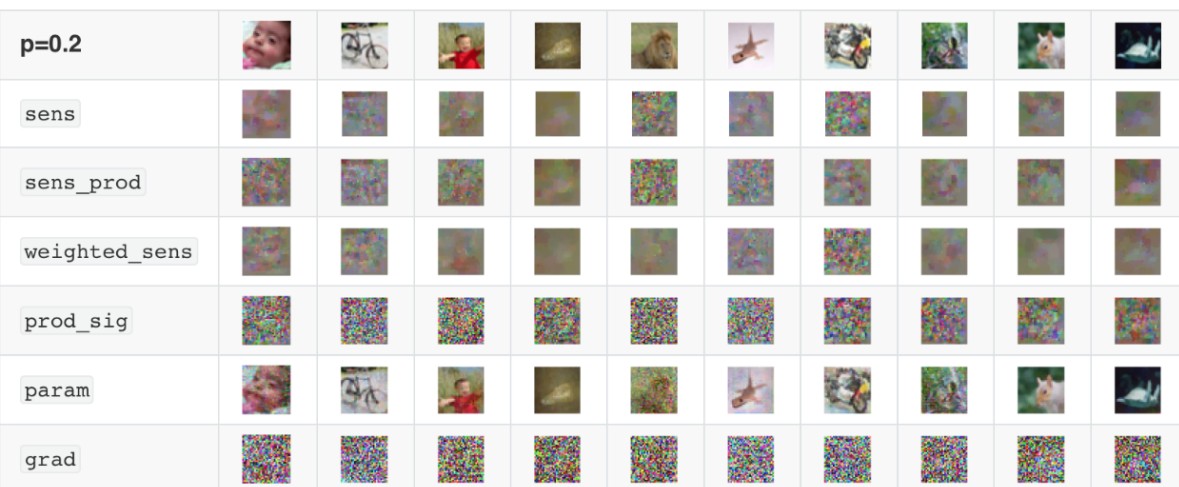

Figure 6: Recovered Images by Inverting Gradients on CNN

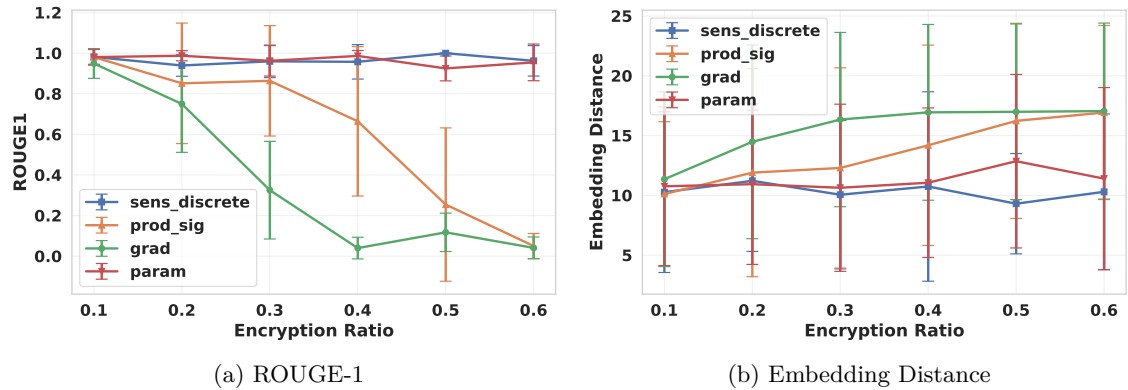

(a) ROUGE-1

(b) Embedding Distance

Figure 7: LAMP on BERT Under Defenses (CoLA)

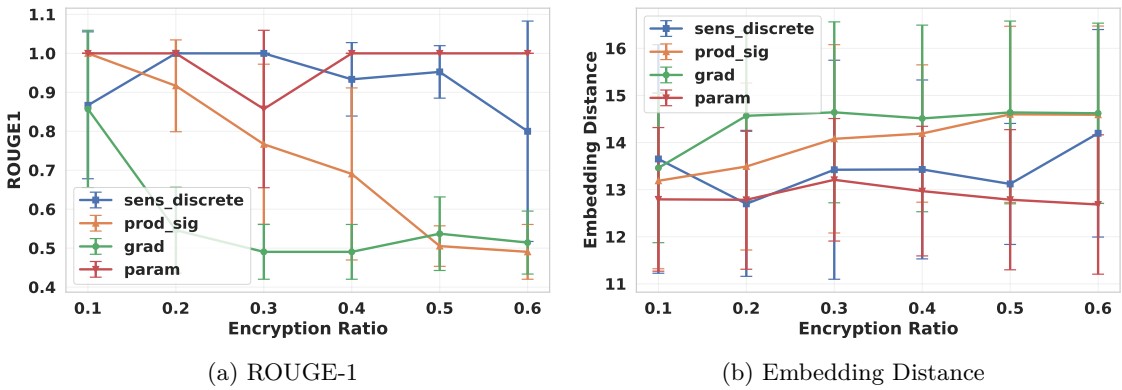

(a) ROUGE-1

(b) Embedding Distance

Figure 8: LAMP on BERT Under Defenses (SST-2)

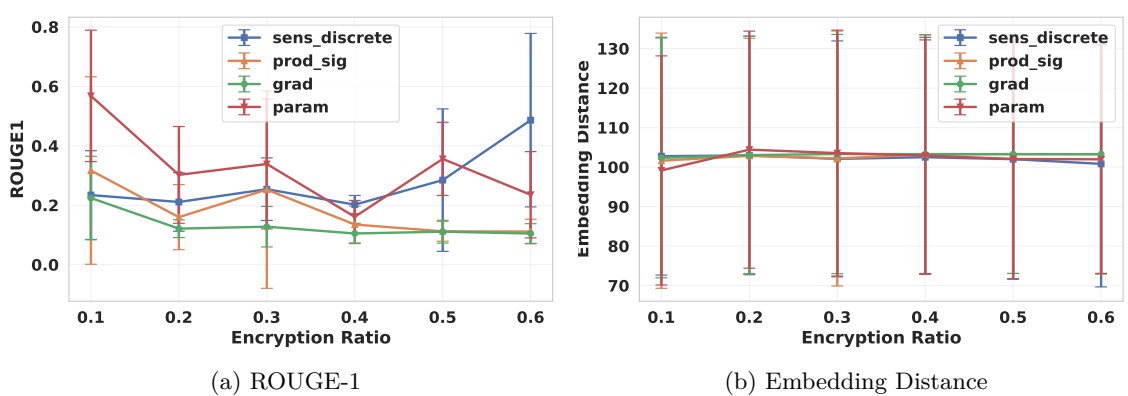

(a) ROUGE-1

(b) Embedding Distance

Figure 9: LAMP on BERT Under Defenses (Rotten Tomatoes)

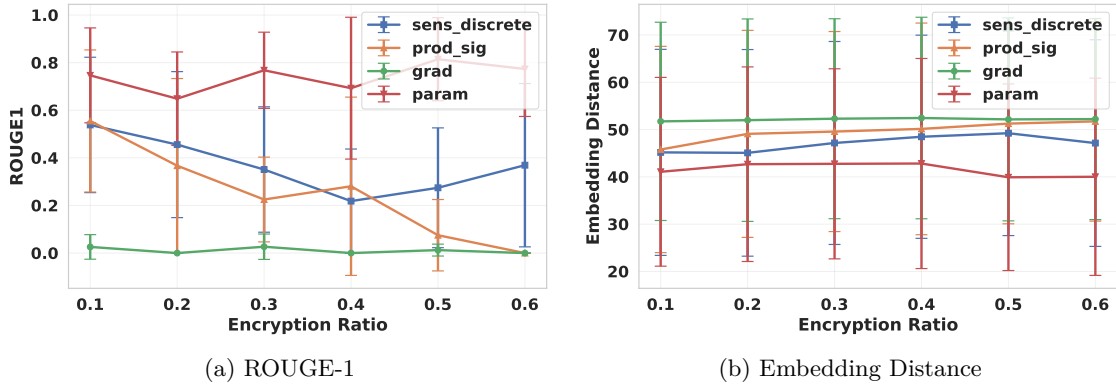

(a) ROUGE-1

(b) Embedding Distance

Figure 10: LAMP on GPT-2 Under Defenses (CoLA)

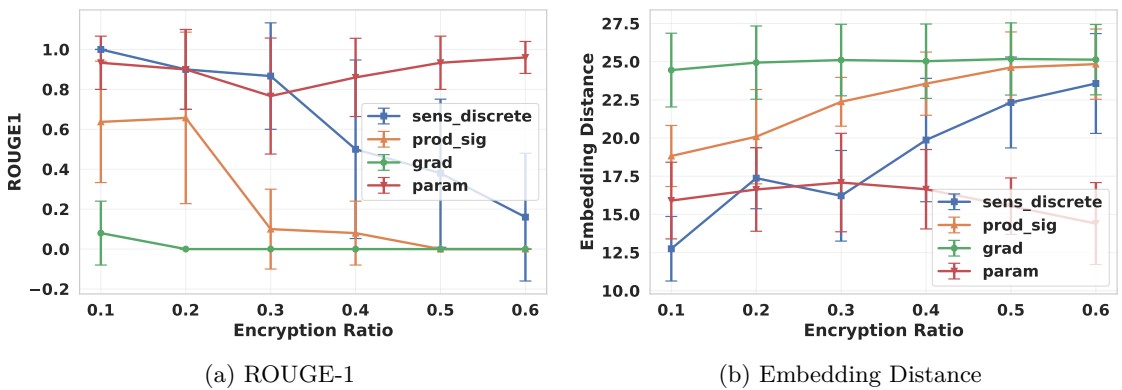

(a) ROUGE-1

(b) Embedding Distance

Figure 11: LAMP on GPT-2 Under Defenses (SST-2)

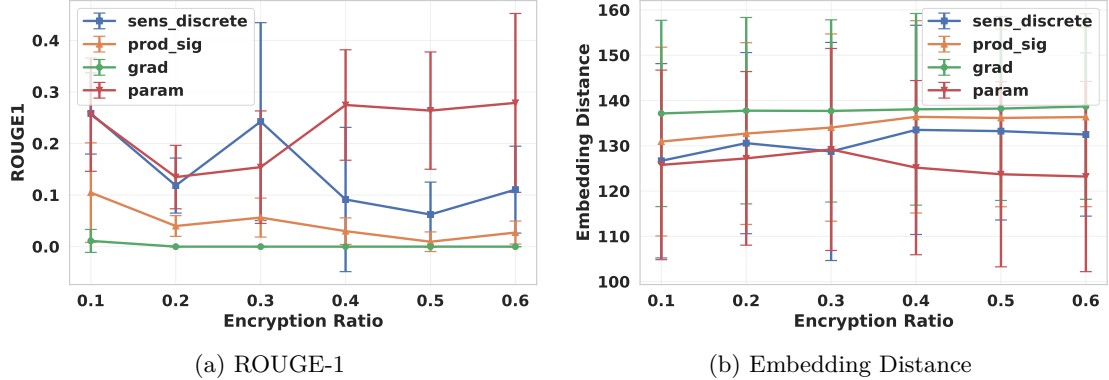

(a) ROUGE-1

(b) Embedding Distance

Figure 12: LAMP on GPT-2 Under Defenses (Rotten Tomatoes)

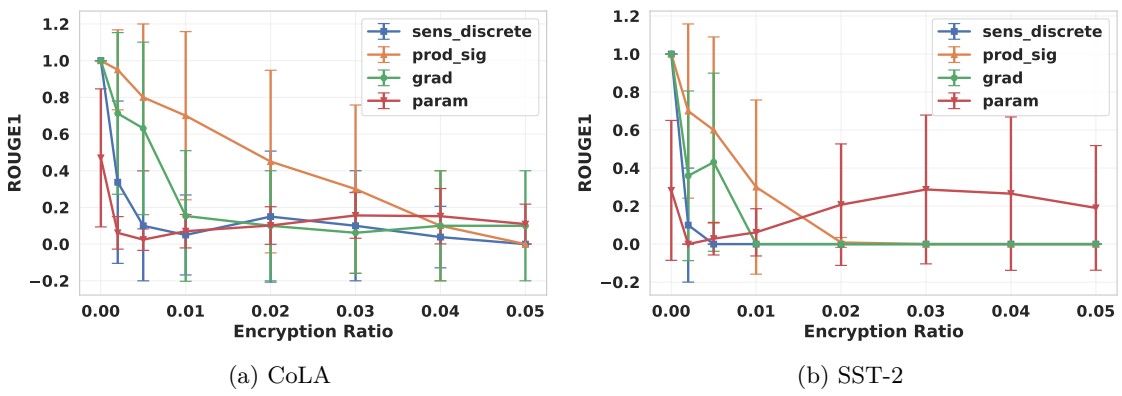

(a) CoLA                          (b) SST-2

Figure 13: DAGER on GPT-2 Under Defenses

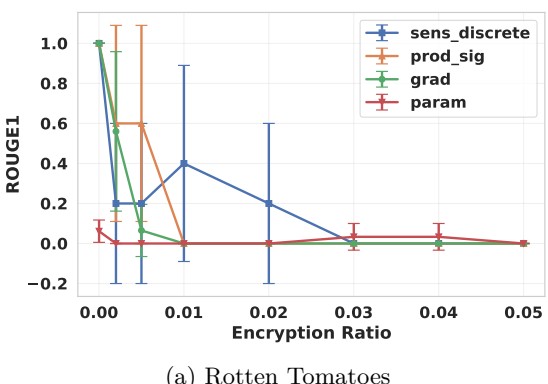

(a) Rotten Tomatoes

Figure 14: DAGER on GPT-2 Under Defenses

## A.5 Reconstruction Loss During Inverting Gradients on Image Models

**LeNet**

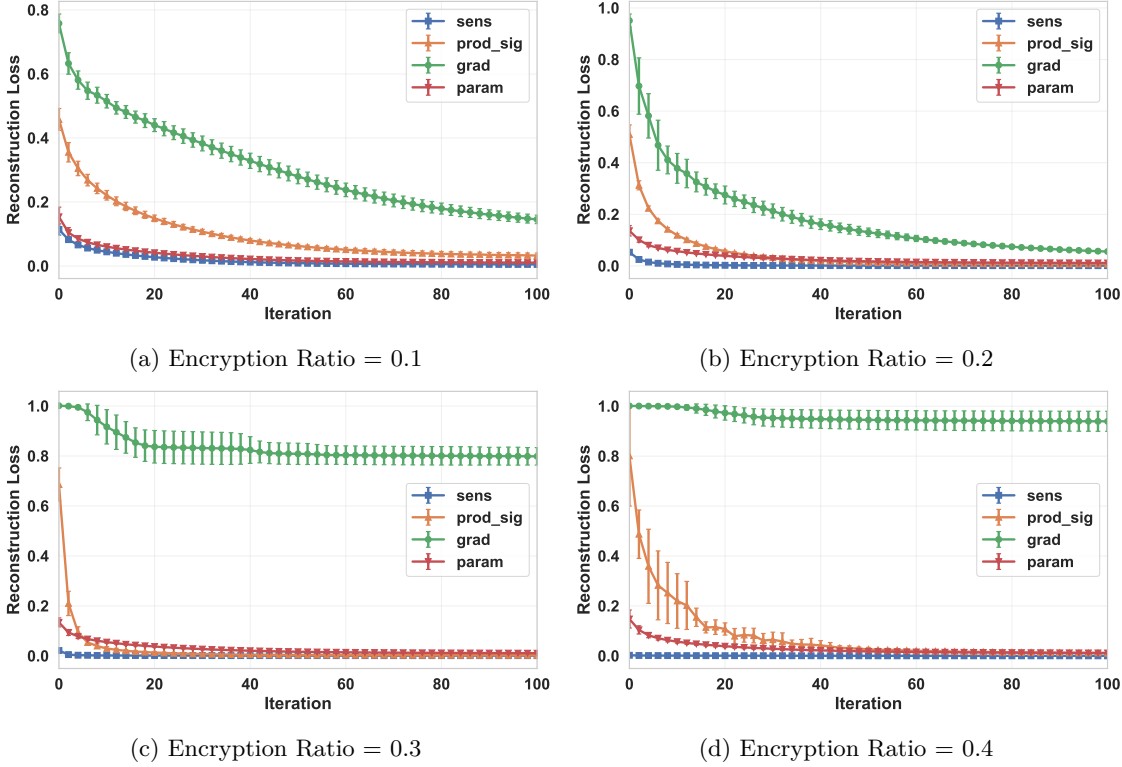

(a) Encryption Ratio = 0.1

(b) Encryption Ratio = 0.2

(c) Encryption Ratio = 0.3

(d) Encryption Ratio = 0.4

Figure 15: Reconstruction Loss During Inverting Gradients on LeNet (Image 2)

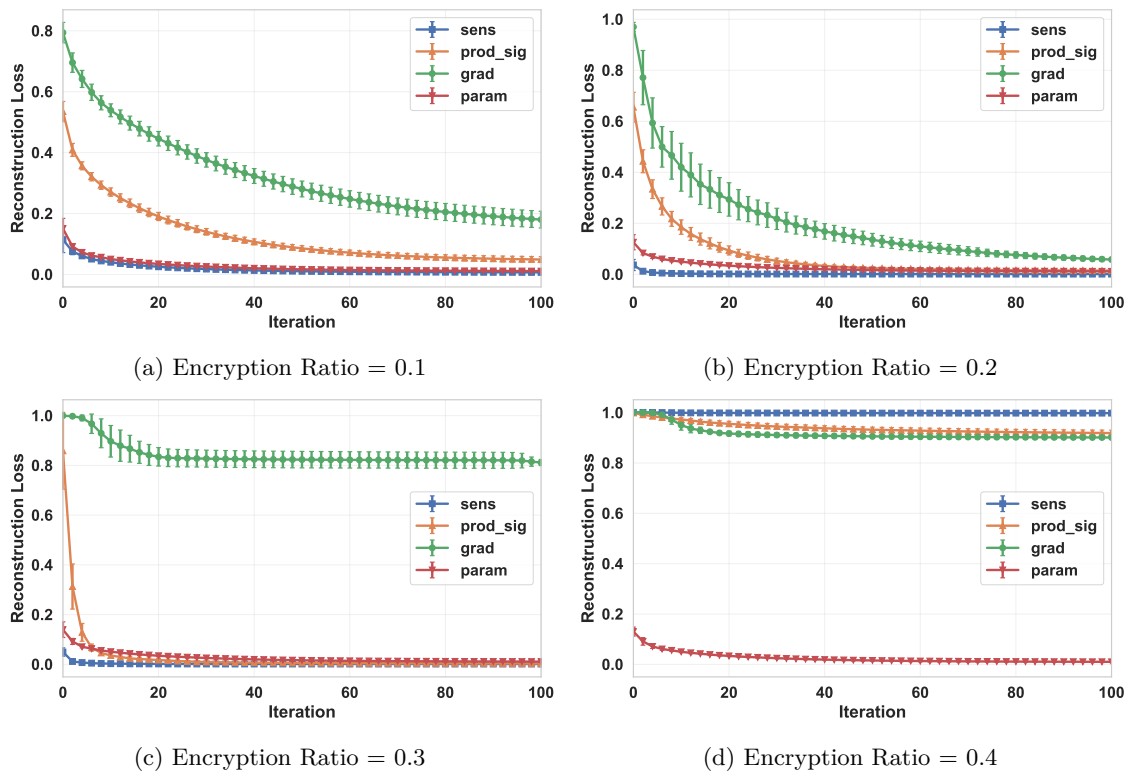

(a) Encryption Ratio = 0.1

(b) Encryption Ratio = 0.2

(c) Encryption Ratio = 0.3

(d) Encryption Ratio = 0.4

Figure 16: Reconstruction Loss During Inverting Gradients on LeNet (Image 3)

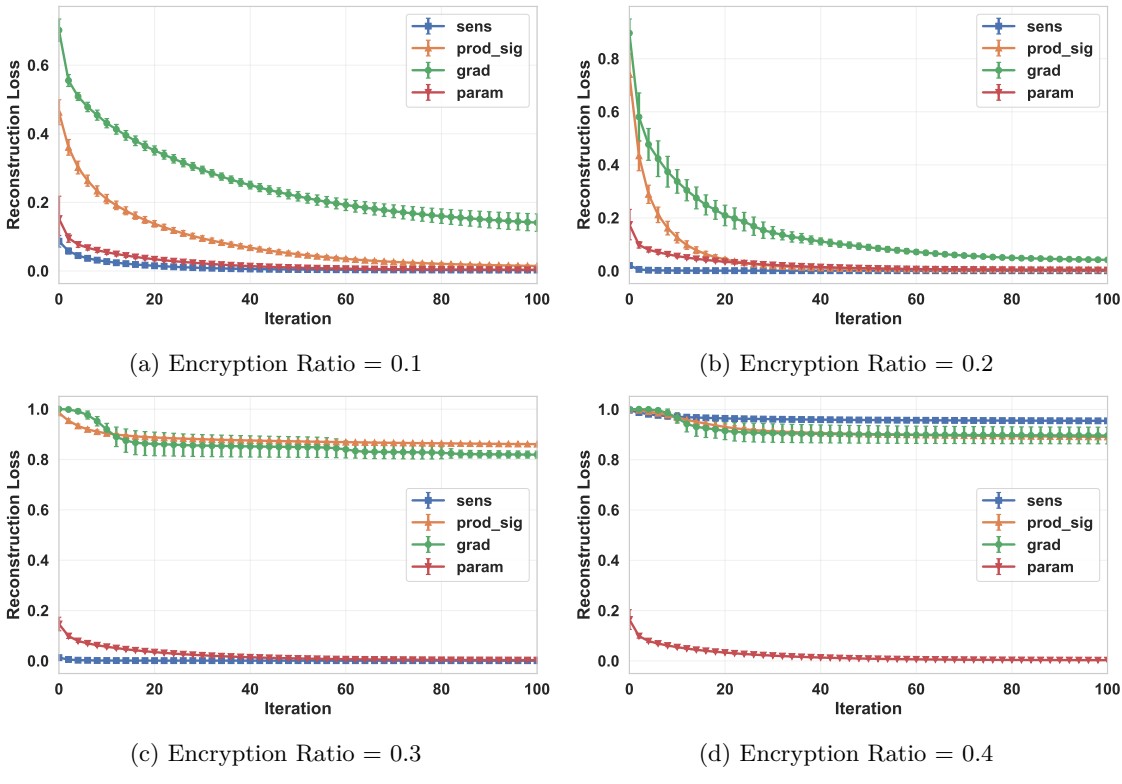

(a) Encryption Ratio = 0.1

(b) Encryption Ratio = 0.2

(c) Encryption Ratio = 0.3

(d) Encryption Ratio = 0.4

Figure 17: Reconstruction Loss During Inverting Gradients on LeNet (Image 4)

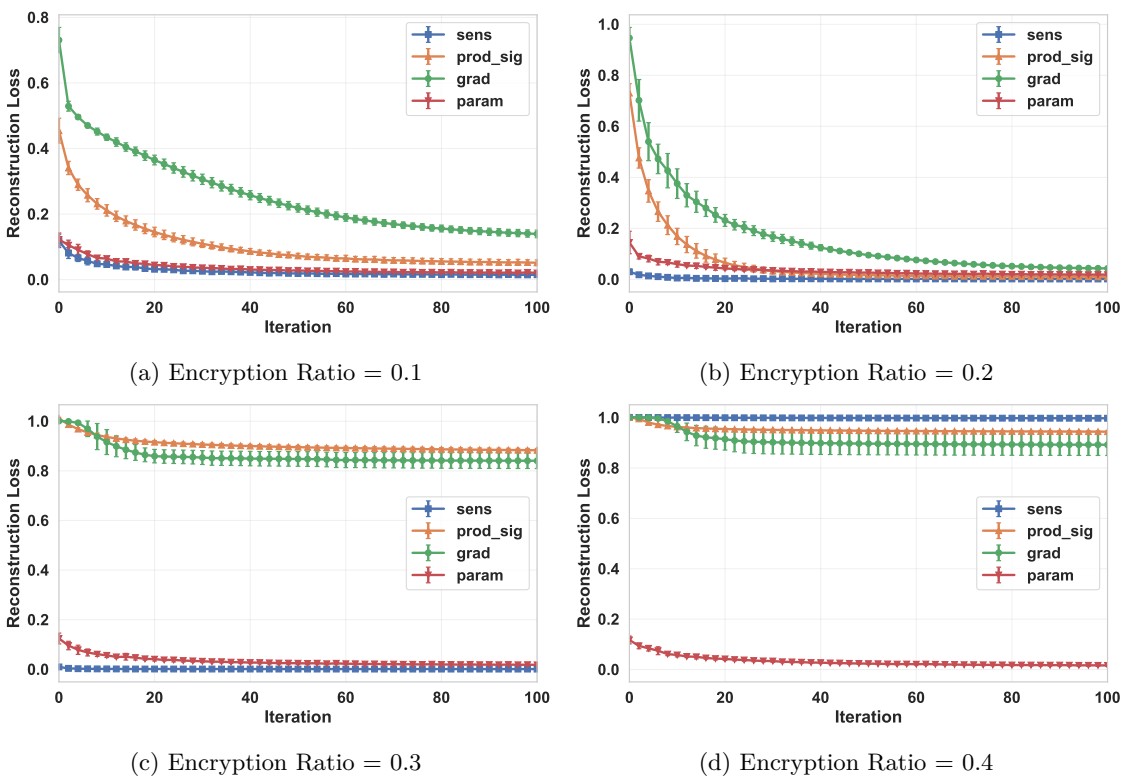

Figure 18: Reconstruction Loss During Inverting Gradients on LeNet (Image 5)

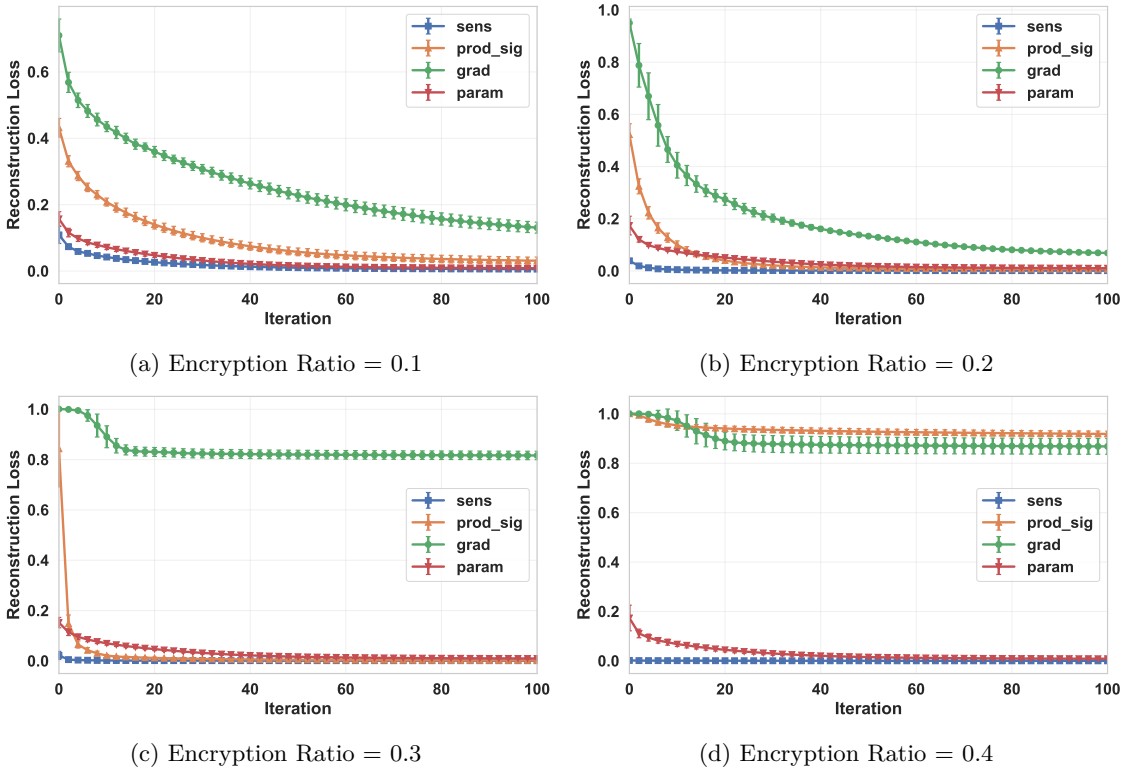

Figure 19: Reconstruction Loss During Inverting Gradients on LeNet (Image 6)

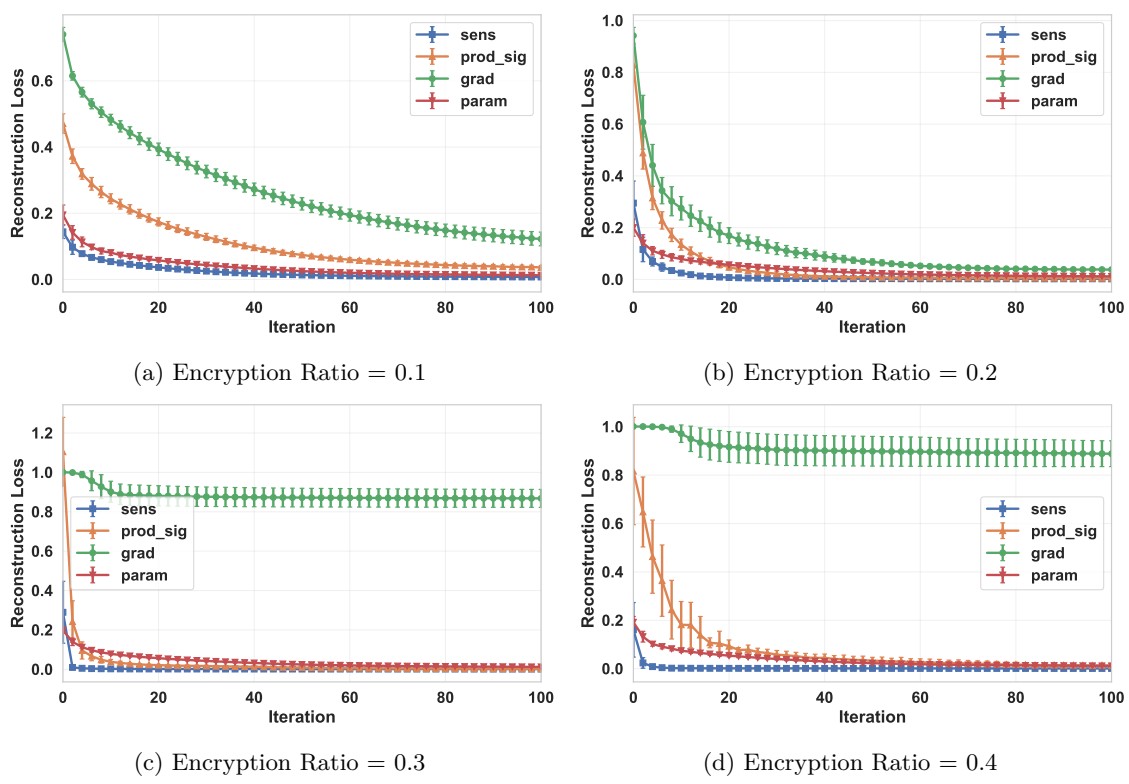

(a) Encryption Ratio = 0.1

(b) Encryption Ratio = 0.2

(c) Encryption Ratio = 0.3

(d) Encryption Ratio = 0.4

Figure 20: Reconstruction Loss During Inverting Gradients on LeNet (Image 7)

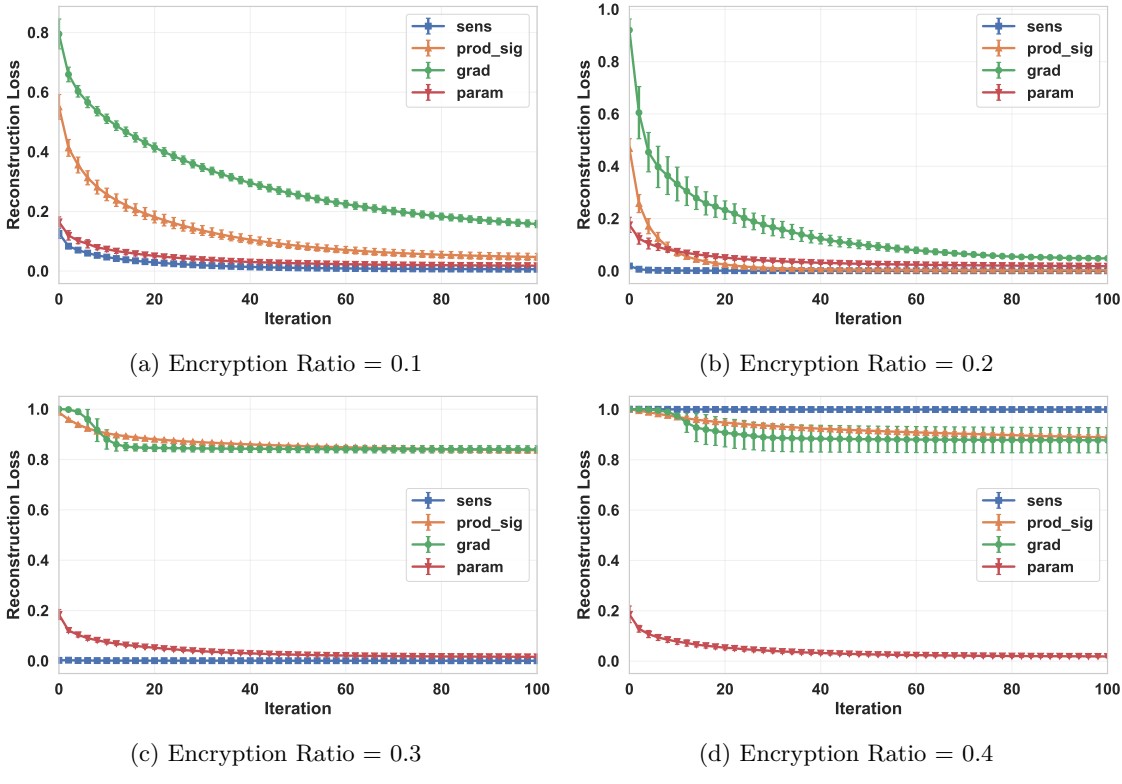

(a) Encryption Ratio = 0.1

(b) Encryption Ratio = 0.2

(c) Encryption Ratio = 0.3

(d) Encryption Ratio = 0.4

Figure 21: Reconstruction Loss During Inverting Gradients on LeNet (Image 8)

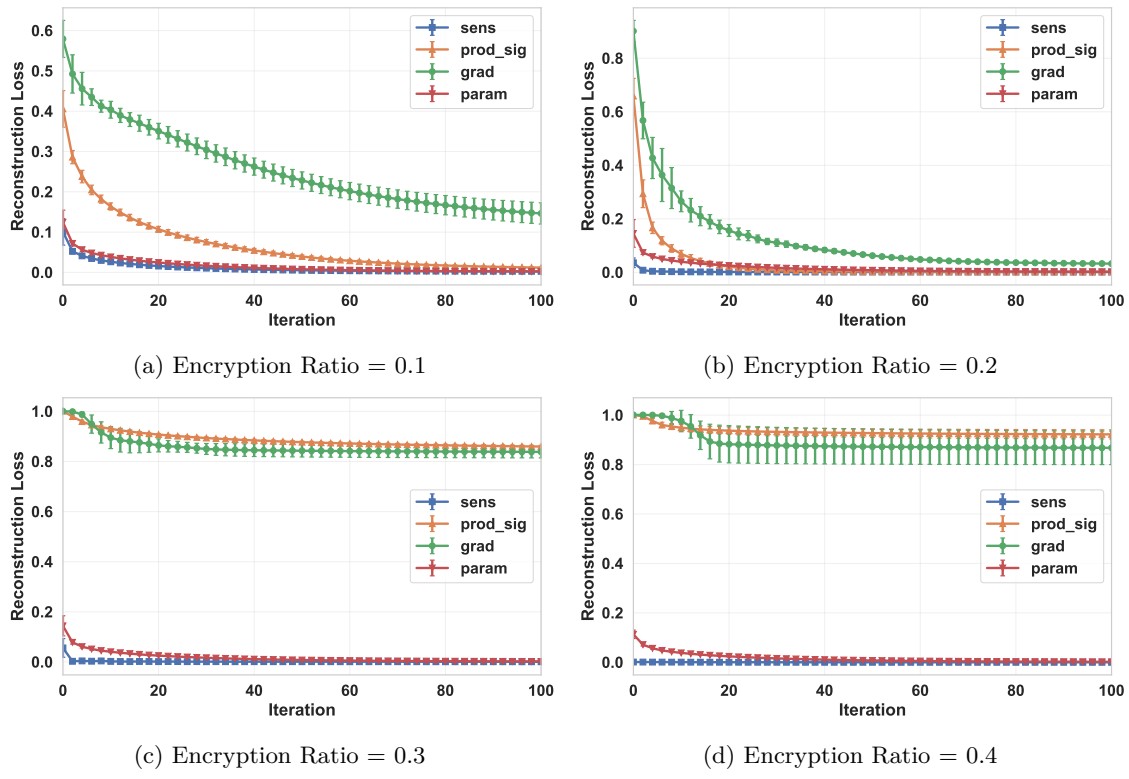

Figure 22: Reconstruction Loss During Inverting Gradients on LeNet (Image 9)

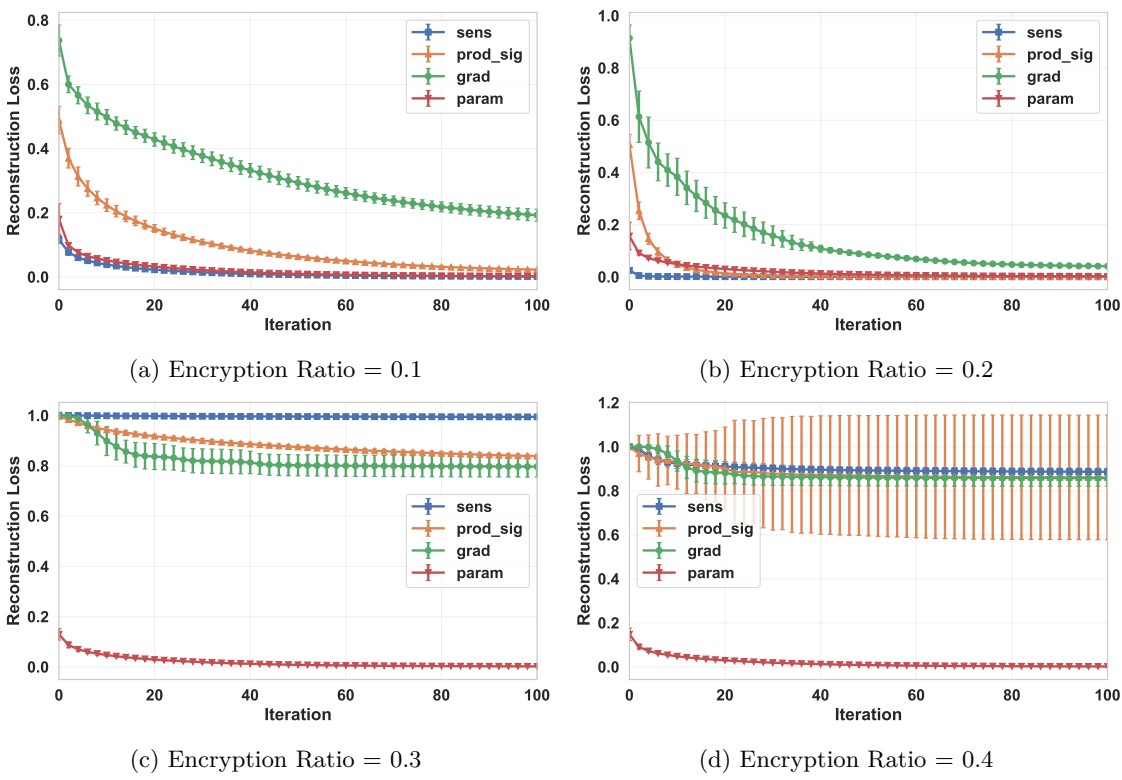

Figure 23: Reconstruction Loss During Inverting Gradients on LeNet (Image 10)

**CNN**

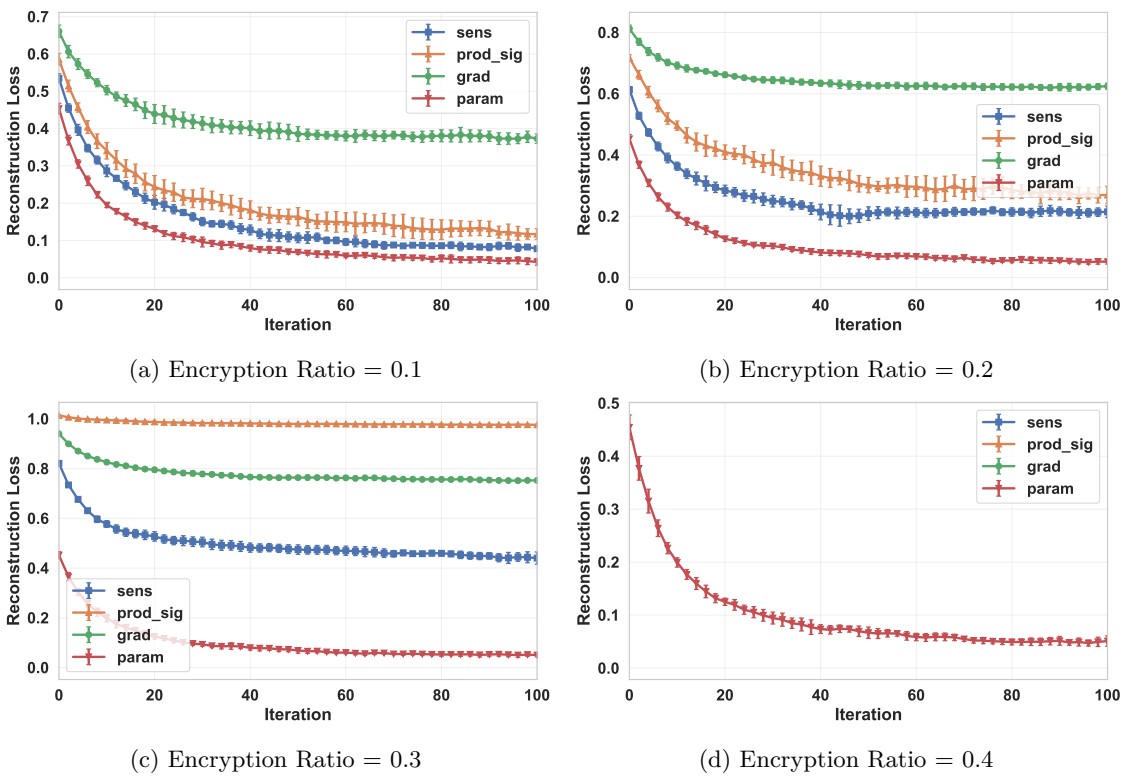

(a) Encryption Ratio = 0.1

(b) Encryption Ratio = 0.2

(c) Encryption Ratio = 0.3

(d) Encryption Ratio = 0.4

Figure 24: Reconstruction Loss During Inverting Gradients on CNN (Image 1)

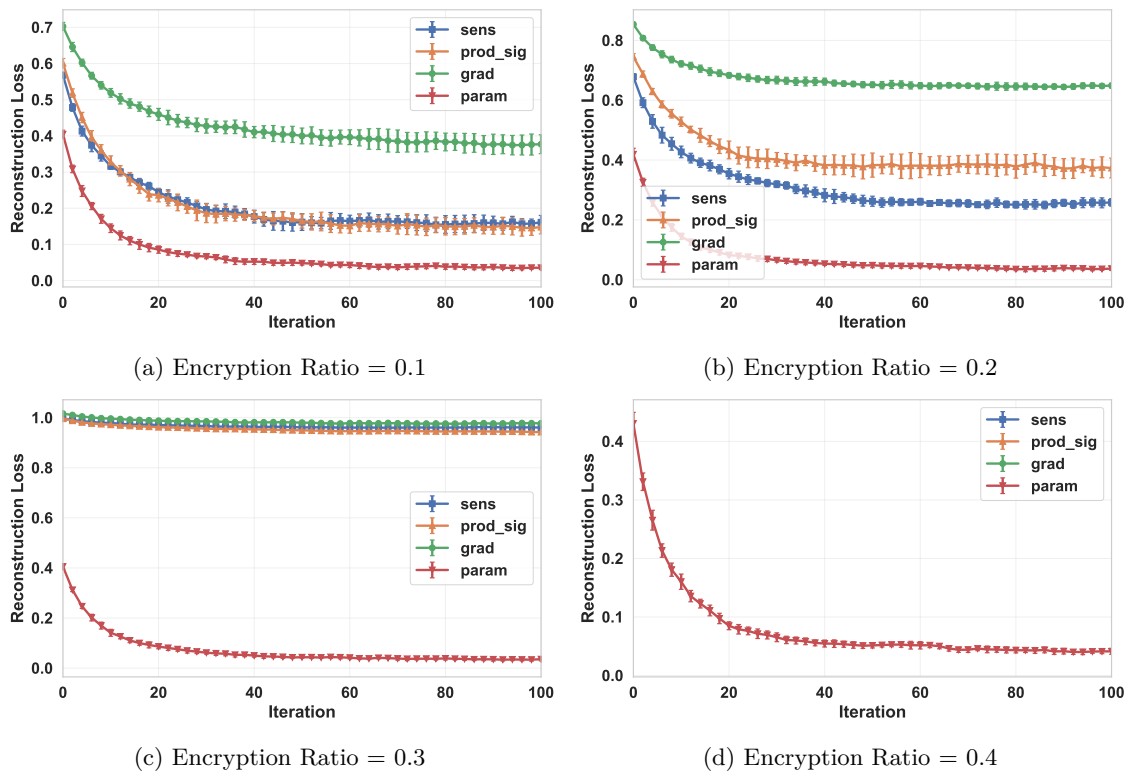

(a) Encryption Ratio = 0.1

(b) Encryption Ratio = 0.2

(c) Encryption Ratio = 0.3

(d) Encryption Ratio = 0.4

Figure 25: Reconstruction Loss During Inverting Gradients on CNN (Image 2)

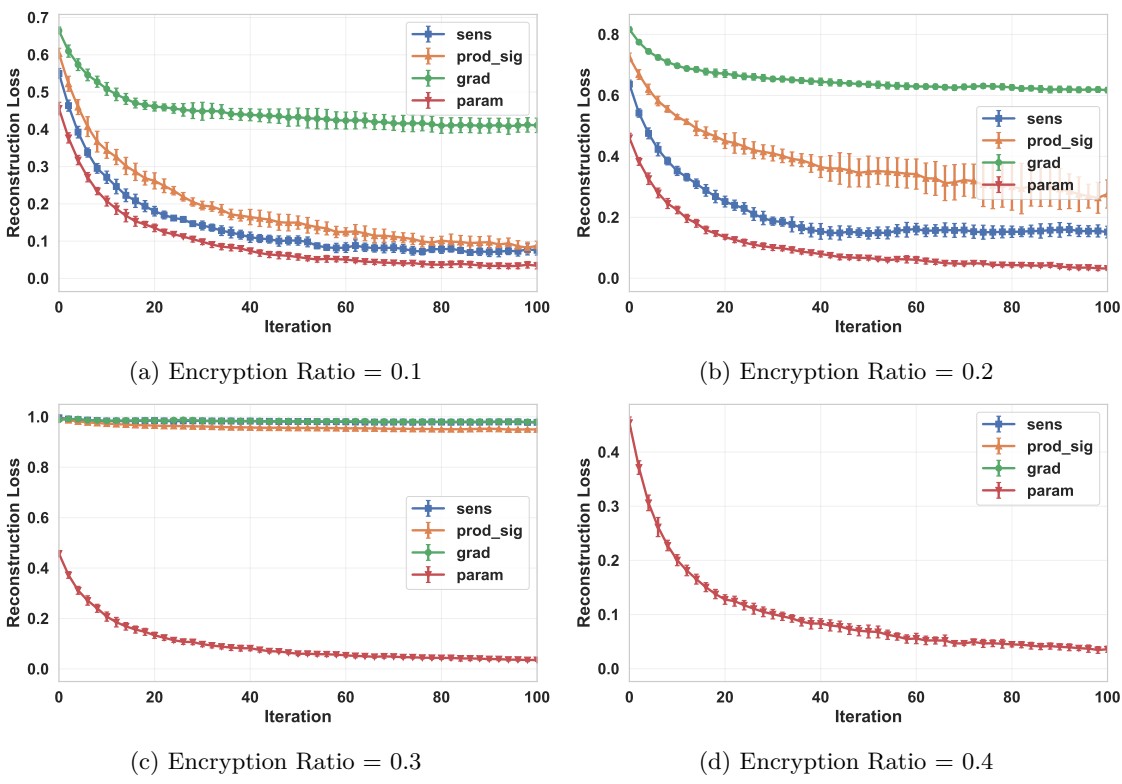

(a) Encryption Ratio = 0.1

(b) Encryption Ratio = 0.2

(c) Encryption Ratio = 0.3

(d) Encryption Ratio = 0.4

Figure 26: Reconstruction Loss During Inverting Gradients on CNN (Image 3)

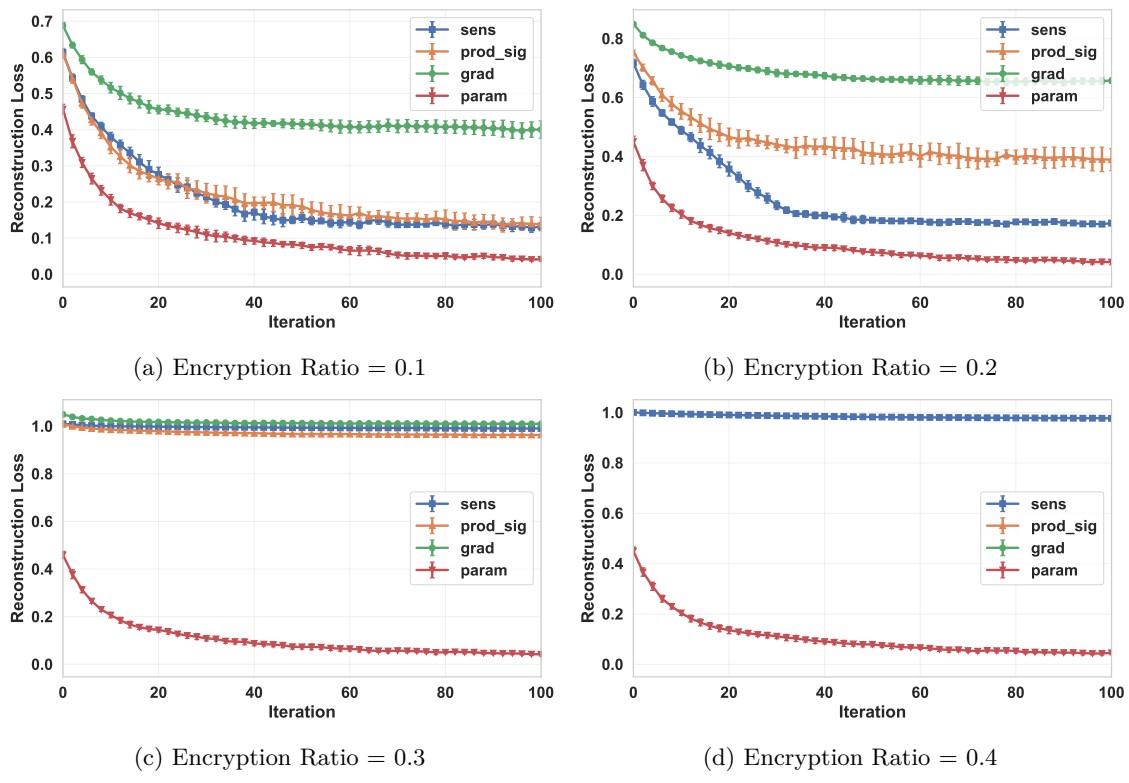

(a) Encryption Ratio = 0.1

(b) Encryption Ratio = 0.2

(c) Encryption Ratio = 0.3

(d) Encryption Ratio = 0.4

Figure 27: Reconstruction Loss During Inverting Gradients on CNN (Image 4)

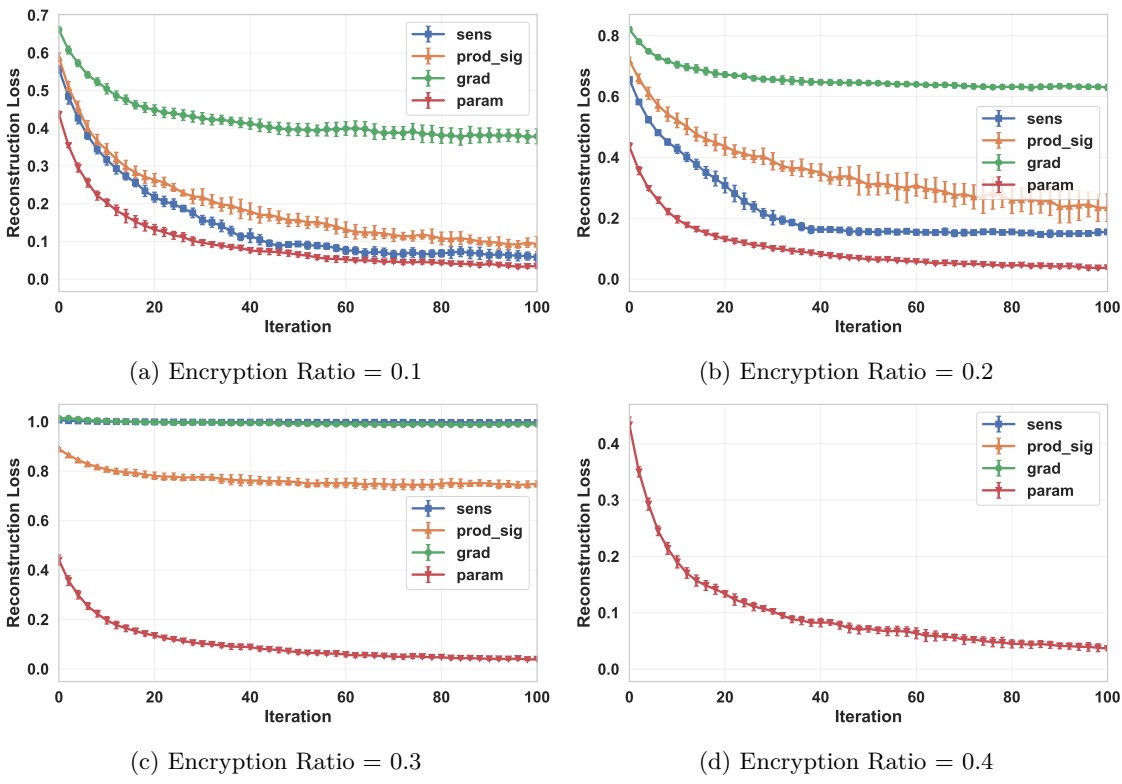

(a) Encryption Ratio = 0.1

(b) Encryption Ratio = 0.2

(c) Encryption Ratio = 0.3

(d) Encryption Ratio = 0.4

Figure 28: Reconstruction Loss During Inverting Gradients on CNN (Image 5)

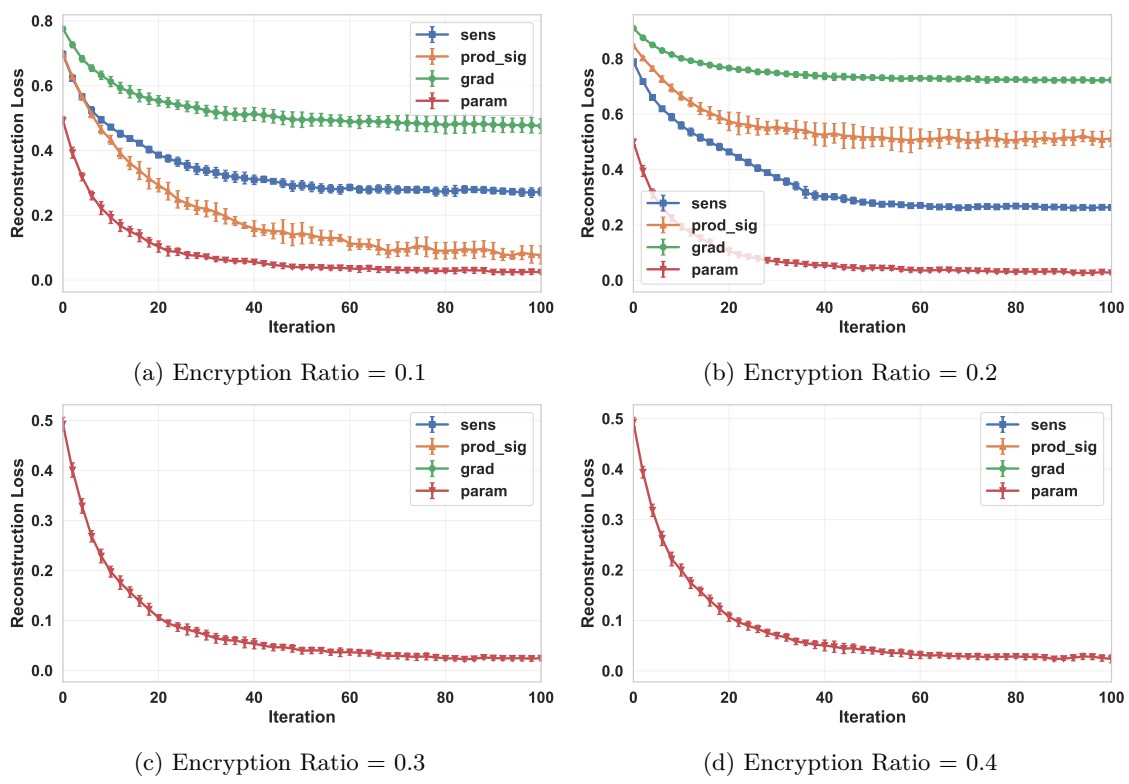

(a) Encryption Ratio = 0.1

(b) Encryption Ratio = 0.2

(c) Encryption Ratio = 0.3

(d) Encryption Ratio = 0.4

Figure 29: Reconstruction Loss During Inverting Gradients on CNN (Image 6)

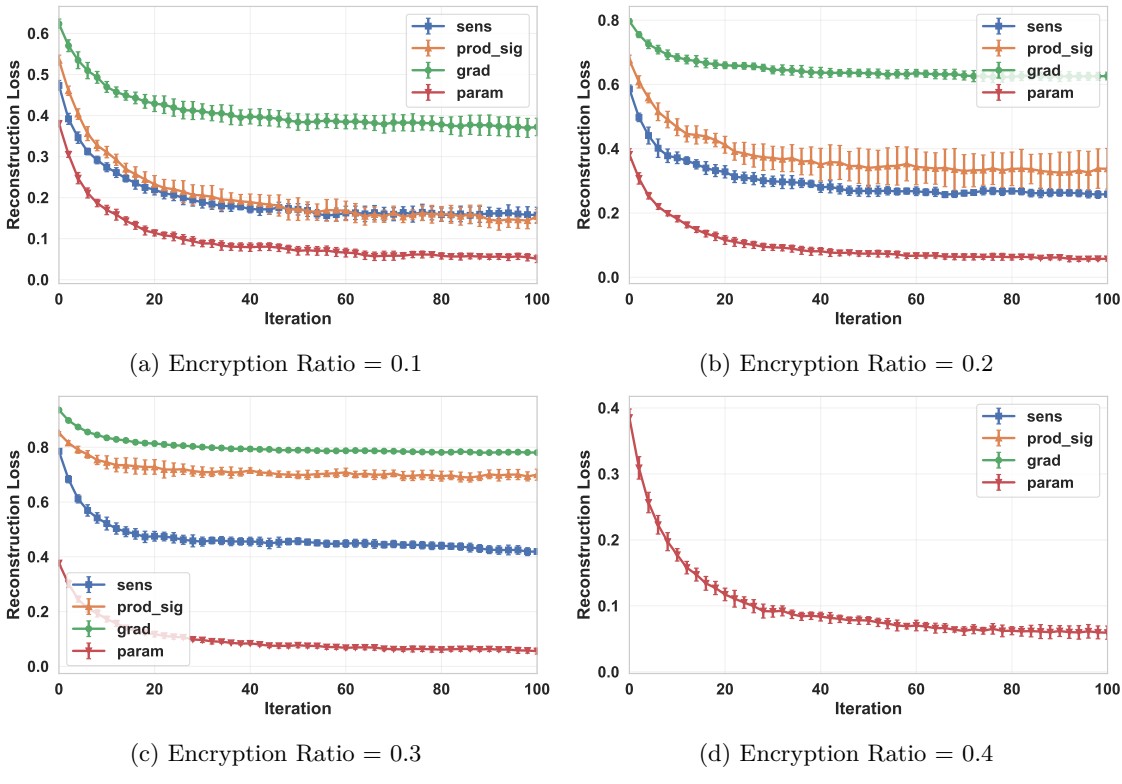

(a) Encryption Ratio = 0.1

(b) Encryption Ratio = 0.2

(c) Encryption Ratio = 0.3

(d) Encryption Ratio = 0.4

Figure 30: Reconstruction Loss During Inverting Gradients on CNN (Image 7)

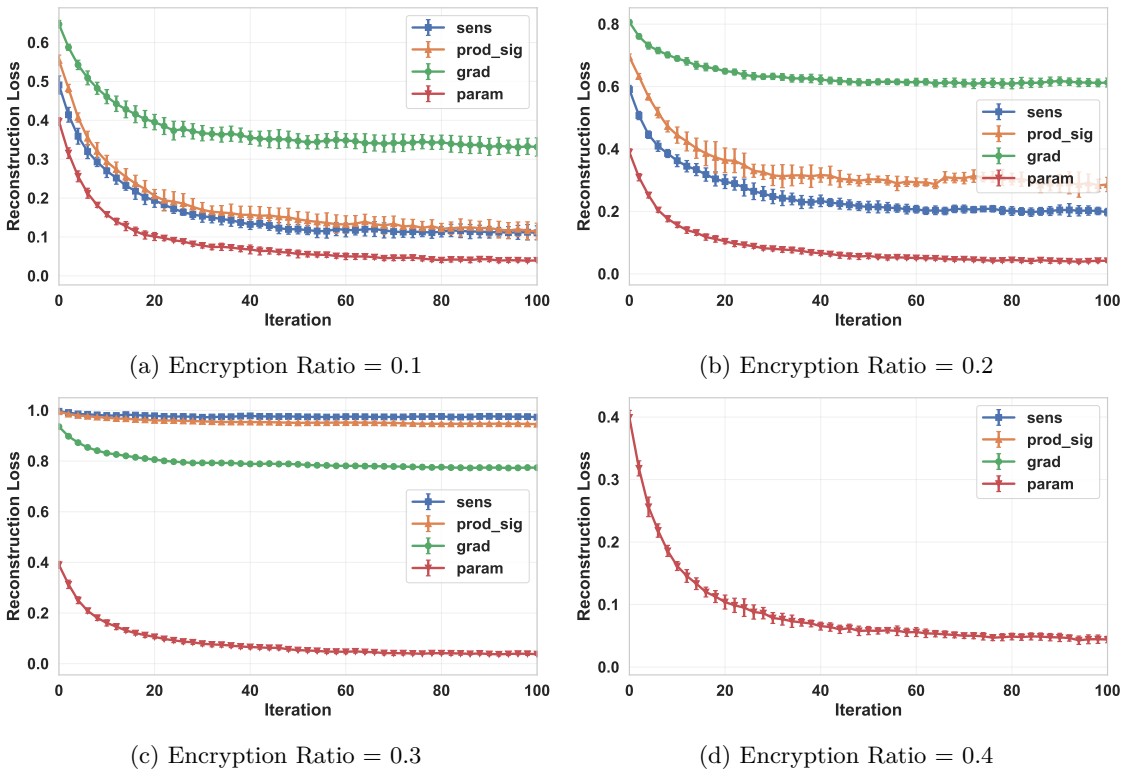

Figure 31: Reconstruction Loss During Inverting Gradients on CNN (Image 8)

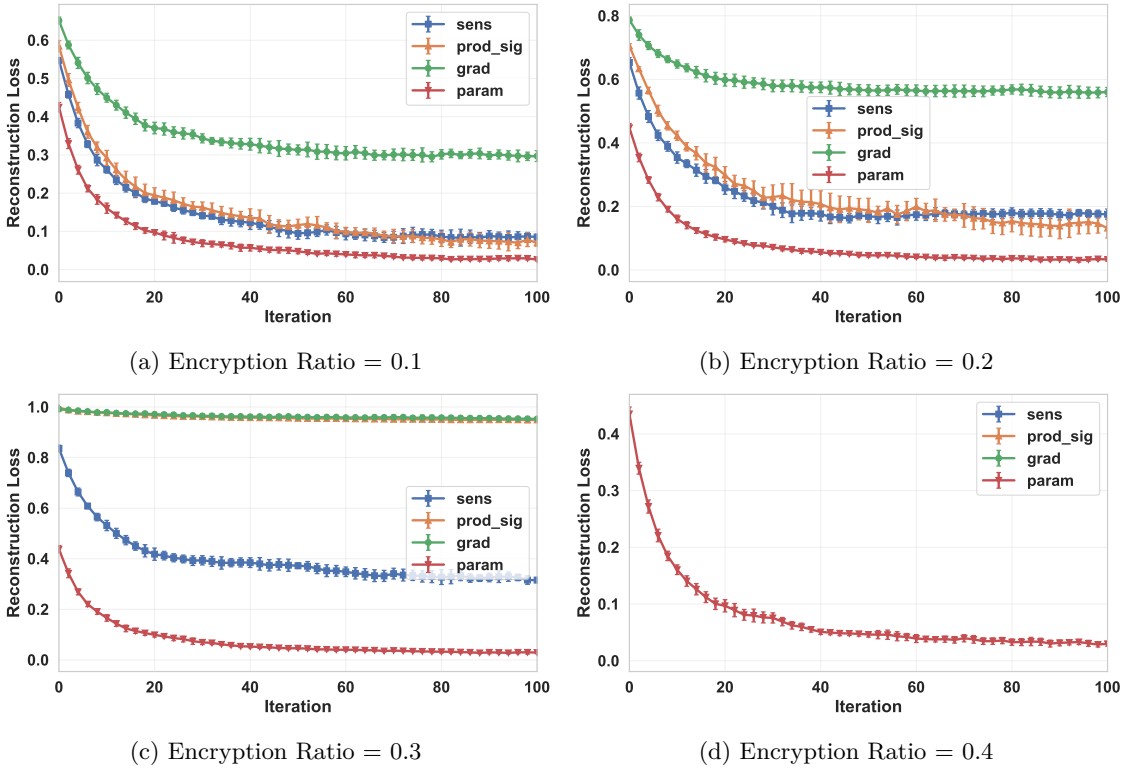

Figure 32: Reconstruction Loss During Inverting Gradients on CNN (Image 9)

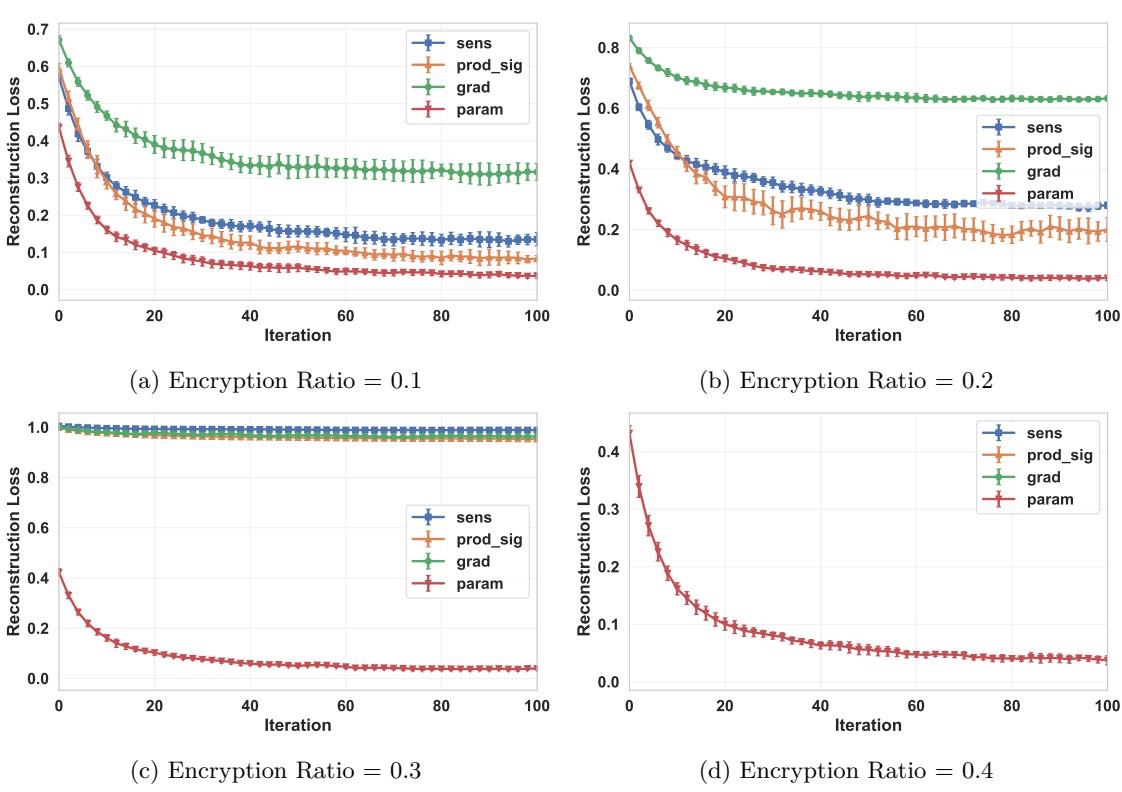

(a) Encryption Ratio = 0.1

(b) Encryption Ratio = 0.2

(c) Encryption Ratio = 0.3

(d) Encryption Ratio = 0.4

Figure 33: Reconstruction Loss During Inverting Gradients on CNN (Image 10)

