# OpenReview forum: "Evaluating Selective Encryption Against Gradient Inversion Attacks"
_TMLR — Rejected by TMLR_

### Review · Reviewer_hhk2 · 2025-07-05

**Summary Of Contributions:**

Gradient inversion attacks enable adversaries to reconstruct private training data from shared model gradients, posing serious privacy threats in federated learning. This paper systematically evaluates selective encryption as a defense mechanism, focusing on encrypting only the most sensitive gradient elements to reduce computational overhead. The authors propose a distance-based significance analysis framework and assess several metrics, such as gradient magnitude, product significance, sensitivity, and parameter magnitude, for identifying critical components. Results show that no single metric is universally optimal, highlighting the need to tailor encryption strategies to specific models and threat scenarios.

**Audience:**

No

**Broader Impact Concerns:**

N/A.

**Claims And Evidence:**

No

**Requested Changes:**

I recommend being clear about the main contribution and the target audience.

**Strengths And Weaknesses:**

To be transparent, I am not an expert in selective encryption, so if any of my comments are based on misunderstandings, I welcome clarification.

1. Unclear Goal and Positioning of Contributions
The goal of the paper is somewhat unclear. The authors initially present selective encryption as a computationally efficient approach for providing strong privacy guarantees in federated learning. However, in the conclusion, the authors state that no single selective encryption strategy is universally optimal across different attack scenarios. This weakens the positioning of their proposed distance-based significance measure, and leaves its practical value somewhat ambiguous. It is not clear what the key takeaway should be for practitioners or researchers.

2. Weak Connection Between Selective Encryption and Gradient Inversion
The paper does not clearly establish the connection between selective encryption techniques and their applicability to defending against gradient inversion attacks. In Section 2.3, for instance, the authors briefly mention prior work such as DCT-based methods and Cai et al. (2019), but do not clearly explain how these methods relate to, or fail to address, gradient inversion threats. From a reader’s perspective, this section reads more like a list of selective encryption techniques rather than a coherent discussion on how these methods might be adapted or extended to the gradient inversion setting. If the paper aims to demonstrate the superiority of selective encryption over alternatives like homomorphic encryption or differential privacy, direct empirical comparisons should be included in the experimental results.

3. Theoretical Claims Lack Strong Empirical Support
The authors claim that their distance-based significance analysis provides a rigorous lower bound on data reconstruction error and that certain metrics (such as gradient magnitude) are computationally efficient and effective. However, the empirical results do not clearly substantiate these claims. While some metrics do perform better in certain cases, the observed gains are not always consistent or substantial enough to convincingly support the claimed theoretical advantages.

4. Limited Experimental Scale
In Section 5.2, the authors mention using only 10 random samples for evaluation. Given the known variability in gradient inversion attack success, it is unclear whether this sample size is sufficient for robust performance assessment. A larger and more diverse set of evaluation samples would strengthen the paper’s conclusions.

---

### Review · Reviewer_up7N · 2025-07-18

**Summary Of Contributions:**

The submission deals with defences against gradient inversion attacks in federated learning based on selective gradient encryption.
In particular, the authors propose to employ significance metrics based on the gradient of the loss with respect to network parameters. Their use is motivated through a series of technical results that, exploiting assumptions on the loss, on the network and on its gradients, relate these significance metrics to the distance between the reconstructed gradient and its ground truth.
The authors present an experimental evaluation across both vision and language benchmarks, demonstrating that the proposed significance metrics (especially the gradient alone) work well on most of the considered setups.

**Audience:**

Yes

**Broader Impact Concerns:**

None.

**Claims And Evidence:**

No

**Requested Changes:**

I would think that the quality of the submission could be improved by:
- presenting the technical results only as motivations for the intuition behind ProdSig and Grad, rather than suggesting that these metrics are "optimal" in some sense. If they are, then this should be further and more rigorously motivated.
- expanding the experimental evaluation to include different batch sizes or at least a larger number of samples per benchmark.
- including DAGER in Table 5 to get further insight on the relatively better performance of Param in this context.

**Strengths And Weaknesses:**

I believe the paper deals with a subject of likely interest to subsets of the TMLR community, and I found the paper to be mostly well-written. The experimental conclusions highlight the effectiveness of a simple and inexpensive solution (Grad), which I would imagine to be fairly useful for most practitioners.

On the other hand, I believe that the significance of the provided technical results is somehow overstated, and that the assumptions underlying them are fairly restrictive. Furthermore, I am not sure the experimental evaluation can be fully claimed to be comprehensive or extensive.
More in detail:
- the employed threat model, under which perfect reconstruction for the non-encrypted entries, does sound to be extremely worst-case.
- some of the assumptions behind the technical results would not hold for ReLU networks, for instance, which are still fairly popular.
- to motivated Grad and ProdSig, it is claimed that the approximate bounds on the distance between the reconstructed signal gradient and the ground truth is dominated by the absolute values of the ground truth gradient. The authors state that this follows from their assumption that the reconstructed gradient is bounded. However, what if the ground truth satisfies the same bounds? More generally, I feel like some of the claims made on the results (e.g., that ProdSig "maximizes the theoretical bound": as far as I understand it is not even an exact bound) should be toned down.
- In spite of the various benchmarks and networks, the number of samples considered in each experiment (10 or 5, I believe) is fairly small, which could perhaps weaken the experimental conclusions.
- Only the setting with batch size = 1 is considered. Are the authors sure that results would generalise beyond this setting?

Finally, the Grad metric, which appears to be the best-performing one on average, is extremely simple.
While I think simple solutions should be preferable, computing the gradient is of course a necessary step for the pre-existing Sens metric. I am not familiar with the related work in the area, but it would be quite suprising if no previous work in the area considered this before, at least as a baseline. Could the authors comment on this?

---

### Review · Reviewer_p8FD · 2025-10-16

**Summary Of Contributions:**

The authors propose a theoretical framework for finding important gradients elements to encrypt to minimize data reconstruction, offer a comprehensive evaluation of different reconstruction attacks across different model architectures, and provide guidelines for which specific defense techniques to use across these different settings and privacy requirements.

**Audience:**

Yes

**Broader Impact Concerns:**

A broader impact section could be warranted given the nature of the paper as specifically utilizing SOTA gradient-based data reconstruction attacks.

**Claims And Evidence:**

Yes

**Requested Changes:**

1. Table 2 is difficult to read, and the information could be better represented using a bar chart, or even simply percentages for encryption required to meet privacy goal. (strengthens work)
2. Include baseline privacy leakage results for each model + dataset. (critical for recommendation)
3. Include baseline utility/model performance results for each model + dataset (critical for recommendation)
4. In the appendix, briefly explain the significance of each figure. (strengthens work)

**Strengths And Weaknesses:**

Strengths:
- The authors do a good job of explaining the significance of privacy risks of gradient-based reconstruction attacks
- Considering a strong attacker with access to direct gradients provides an effective upper bound on potential downstream leakage

Weaknesses:
- I do not see (or at least it is not clear) where baseline reconstruction results, which will be critical to putting the privacy gains from encryption into perspective
- Metrics used for utility and attack success rate need to be clear
- In section 5., the one-fifth-i approach relies on intuition for choosing which model layers are most information-containing. Could the authors provide a citation or an empirical result to support this conclusion?

---

### Decision · Action_Editor_T8S9 · 2025-11-26

**Recommendation:** Reject

**Audience:**

Yes

**Audience Explanation:**

This paper focuses on a mitigation strategy against gradient inversion attacks, which have been gaining traction in the community.

**Claims And Evidence:**

No

**Claims Explanation:**

This paper presents a defensive technique, selective encryption, for mitigating gradient inversion attacks. All three reviewers noted that the proposed defense is not supported by a comprehensive empirical evaluation, and the AE concurs with their assessment. Beyond the reviewer comments, the AE is also concerned about the feasibility of an adaptive adversary who may still be able to recover sensitive gradients with sufficient effort: for example, through the use of calibrated queries or high-quality generative models. Moreover, all the reviewers indicated that they did not receive the author responses, leaving their concerns unaddressed in the final assessments. Thus, the AE recommends rejection.

**Resubmission Of Major Revision:**

The authors may consider submitting a major revision at a later time.